# Synaptojanin cooperates in vivo with endophilin through an unexpected mechanism

Yongming Dong[1], Yueyang Gou[2], Yi Li[2], Yan Liu[1], Jihong Bai[1,3]*

[1]Basic Sciences Division, Fred Hutchinson Cancer Research Center, Seattle, United States; [2]College of Life Science, Sichuan University, Chengdu, China; [3]Department of Biochemistry, University of Washington, Seattle, United States

**Abstract** Synaptojanin and endophilin represent a classic pair of endocytic proteins that exhibit coordinated action during rapid synaptic vesicle endocytosis. Current models suggest that synaptojanin activity is tightly associated with endophilin through high-affinity binding between the synaptojanin proline-rich domain (PRD) and the endophilin SH3 domain. Surprisingly, we find that truncated synaptojanin lacking the PRD domain sustains normal synaptic transmission, indicating that synaptojanin's core function in vivo resides in the remaining two domains that contain phosphoinositide-phosphatase activities: an N-terminal Sac1 phosphatase domain and a 5-phosphatase domain. We further show that the Sac1 domain plays an unexpected role in targeting synaptojanin to synapses. The requirement for Sac1 is bypassed by tethering the synaptojanin 5-phophatase to the endophilin membrane-bending Bin–Amphiphysin–Rvs (BAR) domain. Together, our results uncover an unexpected role for the Sac1 domain in vivo in supporting coincident action between synaptojanin and endophilin at synapses.

*For correspondence: jbai@ fredhutch.org

**Competing interests:** The authors declare that no competing interests exist.

## Introduction

Synaptic vesicle (SV) endocytosis occurs through rapid and coordinated actions of endocytic proteins (*De Camilli and Takei, 1996*; *Dittman and Ryan, 2009*; *Saheki and De Camilli, 2012*). A classic example is the functional pair of synaptojanin and endophilin (*Gad et al., 2000*; *Schuske et al., 2003*; *Song and Zinsmaier, 2003*; *Verstreken et al., 2003*; *Dickman et al., 2005*; *Milosevic et al., 2011*; *Sullivan, 2011*). Synaptojanin is a neuronal phosphoinositide phosphatase that hydrolyzes phosphatidylinositol-4,5-bisphosphate (PI(4,5)P2) to facilitate SV recycling at presynaptic terminals (*McPherson et al., 1996*; *Cremona et al., 1999*; *Harris et al., 2000*; *Verstreken et al., 2003*). Deletion of synaptojanin leads to severe synaptic defects, including depletion of SVs, accumulation of endocytic intermediates, and subsequent failure in synaptic transmission (*Cremona et al., 1999*; *Harris et al., 2000*; *Verstreken et al., 2003*; *Van Epps et al., 2004*; *Dickman et al., 2005*). Overexpression of synaptojanin causes PI(4,5)P2 deficiency and learning deficits in Down syndrome model mice (*Voronov et al., 2008*). While the importance of synaptojanin is well documented, the precise mechanisms for its role in SV recycling remain elusive.

Genetic studies have shown that the function of synaptojanin is tightly linked to the endocytic protein endophilin (*Verstreken et al., 2002*; *Schuske et al., 2003*; *Dickman et al., 2005*). Mutant animals lacking either synaptojanin or endophilin share identical defects at synapses. These defects are not exacerbated in double mutants, supporting that synaptojanin and endophilin function in the same pathway. Current models suggest that synaptojanin is transiently recruited to endocytic sites via direct binding between the endophilin SH3 domain and the synaptojanin proline-rich domain (PRD) (*Schuske et al., 2003*; *Verstreken et al., 2003*; *Milosevic et al., 2011*). In vitro binding assays provide

**eLife digest** Nerve cells called neurons can rapidly carry information around the body. Each neuron forms connections called synapses with several other cells to build networks for information exchange. At most synapses, electrical activity in one neuron results in the release of chemicals called neurotransmitters from storage compartments called synaptic vesicles. The neurotransmitters leave the cell and cross the gap between the two neurons to activate the next cell.

After the neurotransmitters have been released, the synaptic vesicles need to be regenerated via a recycling process called endocytosis. This recycling process is very important for synapses to work properly, but it is not clear exactly how it occurs. Two of the proteins involved are called synaptojanin and endophilin. Synaptojanin is made up of three structural units (or 'domains'), including the proline-rich domain and the Sac1 domain. It has been proposed that interactions between endophilin and the proline-rich domain of synaptojanin are essential for vesicle recycling.

Here, Dong et al. studied nematode worms that carry mutant forms of synaptojanin. The experiments show that the Sac1 domain, but not the proline-rich domain, is required for the synapses to work properly. However, the Sac1 domain is not required if synaptojanin is artificially linked to endophilin.

Dong et al.'s findings suggest that synaptojanin uses its Sac1 domains to work with endophilin. A future challenge will be to understand the details of how this cooperative action occurs.

evidence for a biochemical interaction between PRD and SH3 (*Ringstad et al., 1997*; *de Heuvel et al., 1997*), and blocking PRD-SH3 interactions by peptides induces abnormal accumulation of endocytic intermediates at synapses (*Gad et al., 2000*). However, we recently found that truncated endophilin lacking the SH3 domain has synaptic activity in vivo (*Bai et al., 2010*), suggesting that synaptojanin and endophilin interact through PRD-SH3 independent mechanisms. Alternatively, synaptojanin may be recruited through redundant SH3 harboring proteins, such as amphiphysin (*Micheva et al., 1997*) and intersectin (*Evergren et al., 2007*; *Pechstein et al., 2010*).

Synaptojanin harbors two phosphatase domains in addition to the PRD domain (*McPherson et al., 1996*). The N-terminal Sac1 domain removes the phosphate group on the 3- and 4-position from the inositol (*Guo et al., 1999*; *Krebs et al., 2013*), and the adjacent 5-phosphatase targets the phosphate on the 5-position (*Cremona et al., 1999*; *Chang-Ileto et al., 2011*). This configuration of tandem phosphatases is unique to synaptojanin, as other phosphoinositide phosphatases (e.g., OCRL and SHIP1/2) have single catalytic domains that are linked to protein- or membrane-binding domains such as PH, SH2, and C2 domains (*Pirruccello and De Camilli, 2012*). Interactions through the non-catalytic domains often enhance phosphatase specificity in membrane recognition through coincident detection of multiple targets (*Carlton and Cullen, 2005*). While it is thought that synaptojanin's tandem phosphatase domains act together to degrade multiple types of phosphoinositides at synapses (*Guo et al., 1999*), the precise role of synaptojanin's tandem phosphatase domains is unclear.

Here, we show that the functional core of synaptojanin resides in its tandem phosphatase domains rather than the PRD domain. Our results reveal an unexpected mechanism whereby the Sac1 domain displays a non-catalytic function to support coordinated action between synaptojanin and endophilin at synapses.

## Results

### The synaptojanin PRD is not required for synaptic transmission in vivo

We investigated the requirement of synaptojanin PRD using both behavioral and electrophysiological phenotypes as in vivo assays. In *Caenorhabditis elegans*, the *unc-26* gene encodes a highly conserved synaptojanin homologue with identical domain structure to the mammalian synaptojanin (*Harris et al., 2000*) (*Figure 1A*). Mutant worms lacking *unc-26 synaptojanin* have significantly decreased locomotion rates and largely diminished excitatory postsynaptic currents (EPSCs) at neuromuscular junctions (*Harris et al., 2000*) (*Figure 1B–F* and *Table 1*). Because the density of active zone markers (e.g., RIM/UNC-10) remains unchanged in *unc-26* mutants (*Ch'ng et al., 2008*), reduced EPSC

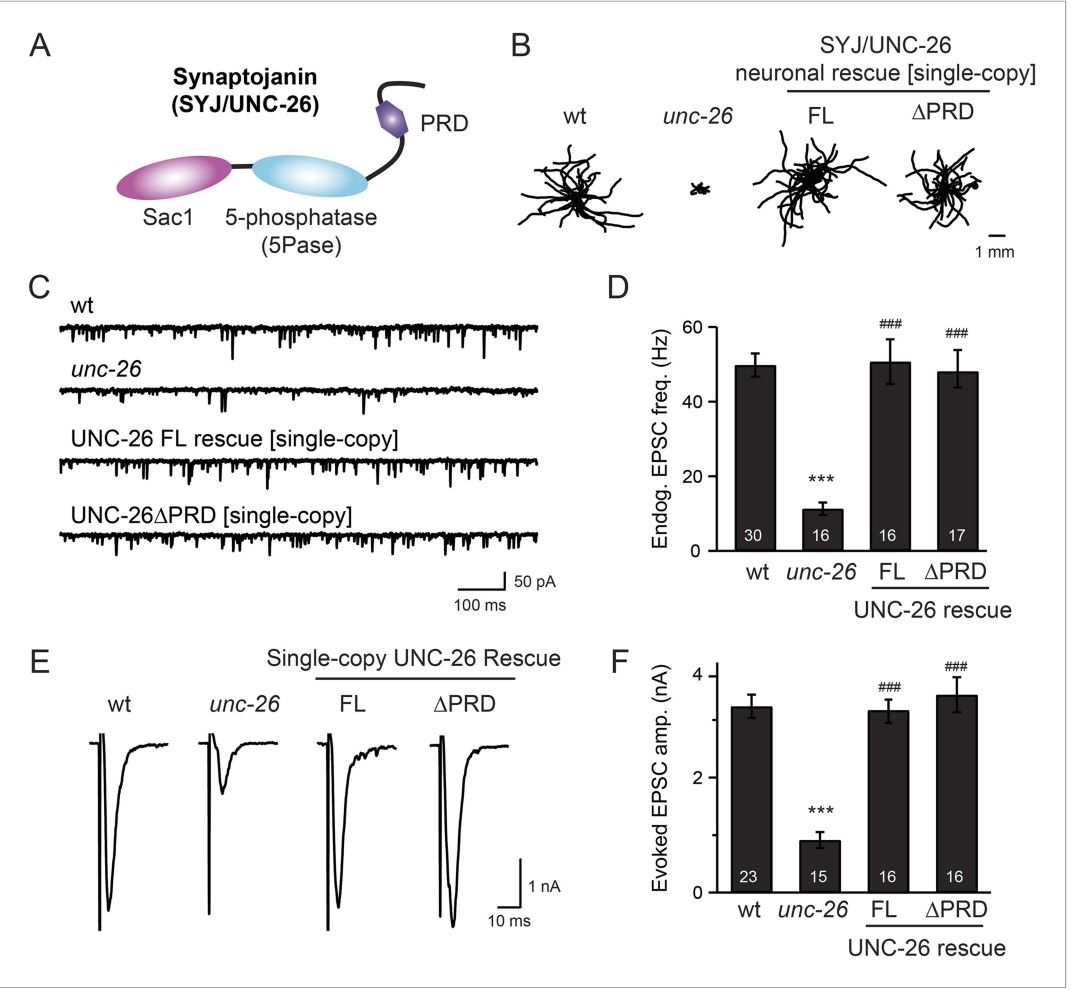

**Figure 1**. Synaptojanin UNC-26 lacking the PRD domain fully supports locomotion, endogenous activity, and evoked synaptic currents. (**A**) Domain structure of synaptojanin UNC-26. Synaptojanin contains three functional modules: a Sac1 phosphatase domain, a 5-phosphatase domain (5Pase), and a proline-rich domain (PRD). Single-copy transgenes encoding GFP-tagged UNC-26 full-length (FL; residues 1–1113) and ΔPRD (residues 1–986) were introduced into *synaptojanin unc-26(s1710)* mutant worms. The pan-neuronal promoter Prab-3 was used to drive transgene expression. (**B**) *C. elegans* locomotion is restored by neuronal expression of full-length synaptojanin (UNC-26FL) or synaptojanin lacking the PRD domain (UNC-26ΔPRD). Representative trajectories (20 animals) of 30 s locomotion are shown for each genotype. The starting points for each trajectory are aligned for clarity.
(**C–F**) Electrophysiological recordings show that GFP-tagged synaptojanin UNC-26ΔPRD is fully functional at synapses. Representative traces and summary data for endogenous EPSC rates (**C–D**) and for evoked EPSC amplitude (**E–F**) are shown for the indicated genotypes. The number of worms analyzed for each genotype is indicated in the bar graphs. ***, p < 0.0001 when compared to wild-type (wt) controls. ###, p < 0.0001 when compared to *unc-26* mutants. Error bars represent standard error of the mean (SEM).
The following figure supplement is available for figure 1:

**Figure supplement 1**. Mouse synaptojanin ΔPRD is functional in *C. elegans* neurons.

---

frequency and amplitude cannot be explained by fewer synapses. Instead, these defects are consistent with previous reports showing reduced SV pools and a corresponding decrease in synaptic transmission due to the cumulative effects of impaired endocytosis over time (*Cremona et al., 1999*; *Harris et al., 2000*; *Verstreken et al., 2003*; *Dickman et al., 2005*).

To determine whether the PRD of synaptojanin is required for endocytosis, we expressed a truncated version of *C. elegans* synaptojanin UNC-26 (residues 1–986; ΔPRD) that lacks PRD in

**Table 1**. Summary of data from electrophysiological recordings and locomotion analyses

| | | Evoked EPSC Amp. (nA) | Endogenous EPSC | | Locomotion speed (μm/s) |
|---|---|---|---|---|---|
| | | | Frequency (Hz) | Amp. (pA) | |
| Wild type (N2) | | 3.2 ± 0.2 (n = 23) | 49.8 ± 2.5 (n = 30) | 22.4 ± 1.0 | 140 ± 8 (n = 65) |
| unc-26 (s1710) | – | 0.9 ± 0.1 (n = 15)† | 12.0 ± 1.5 (n = 16)† | 20.7 ± 0.6 | 28 ± 3 (n = 65)† |
| | Si[Prab-3::unc-26::gfp] | 3.1 ± 0.2 (n = 16)# | 50.8 ± 5.4 (n = 16)# | 23.2 ± 1.4 | 139 ± 8 (n = 60)# |
| | Si[Prab-3::unc-26ΔPRD::gfp] | 3.4 ± 0.3 (n = 16)# | 47.4 ± 5.0 (n = 17)# | 24.8 ± 1.3 | 141 ± 10 (n = 60)# |
| | Ex[Psnb-1::mSYJ1ΔPRD] | 3.1 ± 0.3 (n = 7)# | 49.6 ± 6.1 (n = 11)# | 21.5 ± 0.6 | 143 ± 10 (n = 37)# |
| | Ex[Prab-3::gfp::unc-26(C378S,D380N)] | 3.5 ± 0.2 (n = 10)# | 53.1 ± 6.6 (n = 10) | 23.8 ± 1.3 | 135 ± 9 (n = 60)# |
| | Ex[Prab-3::gfp::unc-26ΔPRD(C378S, D380N)] | 3.2 ± 0.3 (n = 10)# | 50.7 ± 5.7 (n = 10) | 24.2 ± 0.9 | 130 ± 9 (n = 60)# |
| | Ex[Prab-3::gfp::unc-26(D716A)] | 0.8 ± 0.1 (n = 14) | 10.0 ± 1.8 (N = 15) | 20.9 ± 0.8 | 28 ± 3 (n = 60) |
| | Ex[Prab-3::gfp::unc-26ΔSac1] | 1.2 ± 0.2 (n = 10) | 7.2 ± 1.9 (N = 10) | 21.3 ± 1.2 | 31 ± 3 (n = 60) |
| | Ex[Prab-3::unc-26Sac1 + Prab-3::unc-26ΔSac1] | 1.0 ± 0.2 (n = 10) | 16.1 ± 1.8 (n = 10) | 20.4 ± 0.5 | 32 ± 3 (n = 60) |
| | Ex[Prab-3::unc-26Sac1::IntN + Prab-3::IntC::unc-26ΔSac1] | 3.2 ± 0.4 (n = 10)# | 53.1 ± 7.1 (n = 10)# | 24.8 ± 1.2 | 100 ± 7 (n = 60)# |
| | Ex[Prab-3::unc26ΔSac1::rab-3] | 1.0 ± 0.2 (n = 10) | 13.2 ± 2.3 (n = 10) | 20.4 ± 0.9 | |
| | Ex[Prab-3::unc-26ΔSac1::snb-1] | 1.4 ± 0.2 (n = 11) | 17.9 ± 2.6 (n = 11) | 21.1 ± 0.8 | |
| | Ex[Prab-3::bem1PX::unc-26ΔSac1] | 0.8 ± 0.1 (n = 7) | 6.3 ± 0.6 (n = 7) | 18.8 ± 1.2 | |
| | Ex[Prab-3::plc∂PH::unc-26ΔSac1] | 1.0 ± 0.2 (n = 7) | 8.5 ± 1.8 (n = 7) | 18.7 ± 1.1 | |
| | Ex[Prab-3::btkPH::unc-26ΔSac1] | 1.2 ± 0.2 (n = 11) | 11.4 ± 1.2 (n = 11) | 18.4 ± 0.8 | |
| | Ex[Prab-3::apa-2::unc-26ΔSac1] | 1.1 ± 0.1 (n = 11) | 12.8 ± 1.8 (n = 11) | 19.4 ± 0.8 | |
| | Ex[Prab-3::apb-1::unc-26ΔSac1] | 0.9 ± 0.1 (n = 11) | 13.2 ± 2.5 (n = 11) | 18.2 ± 1.5 | |
| | Ex[Prab-3::apm-2::unc-26ΔSac1] | 1.1 ± 0.1 (n = 10) | 15.3 ± 2.7 (n = 10) | 21.0 ± 0.9 | |
| | Ex[Prab-3::aps-2::unc-26ΔSac1] | 1.3 ± 0.2 (n = 9) | 12.7 ± 1.4 (n = 9) | 20.1 ± 0.9 | |
| | Ex[Prab-3::unc-57::unc-26ΔSac1] | 2.5 ± 0.3 (n = 11)# | 28.0 ± 4.1 (n = 11)§ | 23.0 ± 1.4 | |
| | Ex[Prab-3::dyn-1::unc-26ΔSac1] | 1.3 ± 0.3 (n = 10) | 14.4 ± 2.9 (n = 10) | 19.4 ± 0.9 | |
| | Ex[Prab-3::itsn-1::unc-26ΔSac1] | 0.9 ± 0.1 (n = 9) | 13.9 ± 1.4 (n = 9) | 21.5 ± 1.4 | |
| | Ex[Prab-3::unc-57::unc-26ΔSac1ΔPRD] | 3.0 ± 0.3 (n = 11)# | 36.7 ± 5.4 (n = 11)§ | 21.7 ± 1.3 | |
| | Ex[Prab-3::unc-57::unc-26ΔSac1 (D716A)] | 0.8 ± 0.2 (n = 9) | 13.0 ± 2.3 (n = 9) | 21.0 ± 0.7 | |
| | Ex[Prab-3::unc-57BAR::unc-26ΔSac1] | 2.9 ± 0.2 (n = 12)# | 23.9 ± 2.5 (n = 12)# | 21.3 ± 0.8 | |
| | Ex[Prab-3::rEndoBAR::unc-26ΔSac1] | 3.1 ± 0.4 (n = 13)# | 28.5 ± 4.6 (n = 13)# | 24.1 ± 1.4 | |
| | Ex[Prab-3::mAmphBAR::unc-26ΔSac1] | 1.4 ± 0.2 (n = 10) | 19.2 ± 2.3 (n = 10) | 20.8 ± 0.8 | |
| | Ex[Prab-3::mNadrin2BAR::unc-26ΔSac1] | 1.7 ± 0.2 (n = 11)‡ | 16.1 ± 3.1 (n = 11) | 23.8 ± 1.9 | |
| | Ex[Prab-3::rEndoBARΔN::unc-26ΔSac1] | 1.3 ± 0.2 (n = 12) | 9.1 ± 0.9 (n = 12) | 18.9 ± 0.4 | |
| | Ex[Prab-3::rEndoBAR(K76E,K78E)::unc-26ΔSac1] | 1.5 ± 0.2 (n = 10) | 13.5 ± 1.3 (n = 10) | 19.3 ± 0.7 | |
| N2 | Prab-3::unc-26ΔPRD(D716A) overexpression | 1.5 ± 0.3 (n = 9)* | 24.6 ± 3.8 (n = 10)* | 25.7 ± 0.9 | |
| | Prab-3::unc-26ΔPRD overexpression | 2.9 ± 0.3 (n = 9) | 53.5 ± 4.6 (n = 9) | 25.4 ± 1.3 | |
| | Prab-3::unc-26ΔSac1ΔPRD(D716A) overexpression | 3.5 ± 0.3 (n = 10) | 49.0 ± 7.5 (n = 10) | 25.9 ± 1.9 | |

*Table 1. Continued on next page*

*Table 1. Continued*

| | | Evoked EPSC Amp. (nA) | Endogenous EPSC | | | Locomotion speed (µm/s) |
| | | | Frequency (Hz) | Amp. (pA) | | |
|---|---|---|---|---|---|---|
| unc-57(e406); unc-26 (s1710) | – | 0.8 ± 0.2 (n = 9)† | 8.6 ± 0.8 (n = 10)† | 21.9 ± 1.1 | 27 ± 3 (n = 60)† | |
| | Si[Psnb-1::unc-57ΔSH3::mCherry]; Si[Prab-3::unc-26ΔPRD::gfp] | 3.2 ± 0.2 (n = 9)¶ | 50.3 ± 4.1 (n = 9)¶ | 23.1 ± 1.0 | 142 ± 9 (n = 62)¶ | |
| | Si[Psnb-1::rEndoBAR::unc-26ΔSac1ΔPRD] | 3.0 ± 0.3 (n = 10)¶ | 50.9 ± 4.1 (n = 10)¶ | 26.5 ± 1.1 | 109 ± 4 (n = 68)¶ | |

*p < 0.001 when compared with N2.

†p < 0.0001 when compared with N2.

‡p < 0.05 when compared with *unc-26* mutant.

§p < 0.001 when compared with *unc-26* mutant.

#p < 0.0001 when compared with *unc-26* mutant.

¶p < 0.0001 when compared with *unc-57; unc-26* double mutants.

Si: single-copy transgene (MosSci insertion).

Ex: extrachromosomal array.

'Amp.' indicates amplitude.

*unc-26 null* mutant worms. In transgenic animals, a single copy of the transgene (*unc-26ΔPRD::gfp*) driven by a pan-neuronal promoter (*Prab-3*) was inserted into chromosome X to avoid confounding issues of overexpression (*Frøkjaer-Jensen et al., 2012*). We reasoned that if the PRD domain is essential, the truncated UNC-26ΔPRD should not rescue mutant defects. Surprisingly, similar to full-length UNC-26, UNC-26ΔPRD fully restored locomotion, endogenous EPSCs, and evoked responses in *unc-26* mutant worms (*Figure 1* and *Table 1*). To test the functional conservation between vertebrate and nematode synaptojanin, we expressed a truncated version of mouse synaptojanin 1 (mSyj1ΔPRD, residues 1–1045) in *unc-26* mutants. We found that truncated mSyj1ΔPRD also restored locomotion and synaptic transmission to wild type (wt) levels (*Figure 1—figure supplement 1* and *Table 1*), indicating that synaptojanin from both invertebrate and vertebrate animals remains largely active in the absence of PRD.

To assay for membrane recycling, we employed FM4-64, a fluorescent lipophilic dye that is internalized by endocytosis (*Betz et al., 1996*; *Kay et al., 1999*). In wt animals, dye was readily internalized in response to KCl stimulation, evident by the high level of FM4-64 fluorescence (3527 ± 412 arbitrary units [a.u.]; n = 12) in the neuron ganglion after washing (*Figure 2A–B*). Approximately 43% of internalized FM4-64 (1411 ± 150 a.u.; n = 12) was released after KCl stimulation, indicating that FM4-64 was internalized into recycling vesicles. By contrast, the dye uptake in *unc-26* mutant worms was significantly lower: reduced by ~40% compared to controls (2123 ± 172 a.u.; n = 11), consistent with defects in membrane recycling. About 32% of internalized dye (686 ± 117 a.u.; n = 11) by the *unc-26* mutants was released upon KCl challenge (*Figure 2A–B*). Expression of the single-copy *Prab-3::unc-26ΔPRD::gfp* transgene fully restored FM4-64 uptake (3885 ± 505 a.u.; n = 10) and the KCl-dependent dye release (1569 ± 243 a.u.; n = 10) (*Figure 2A–B*), indicating that the recovery of vesicle recycling processes does not require UNC-26PRD.

We next asked if synapses rescued by truncated UNC-26ΔPRD sustain synaptic transmission upon repetitive stimuli. Cholinergic neurons of transgenic animals carrying *Punc-17::ChR2(H134R)::YFP* were activated by 2-Hz photostimulation, and evoked EPSCs were recorded at neuromuscular junctions (*Liewald et al., 2008*; *Liu et al., 2009*). For all successive stimuli, the amplitudes of EPSCs in *unc-26* mutant worms were significantly reduced compared to those in control worms (*Figure 2C–D*). These results are consistent with previous findings showing that *unc-26* mutant synapses exhibit more depression in synaptic transmission after repeated stimulation, due to impaired endocytosis. Expression of the single-copy *Prab-3::unc-26ΔPRD::gfp* transgene recovered EPSC amplitudes of successive stimuli, supporting the notion that truncated UNC-26 functions sufficiently to supply SVs during sustained activity. Together, these results argue against an essential role of the synaptojanin PRD domain at synapses.

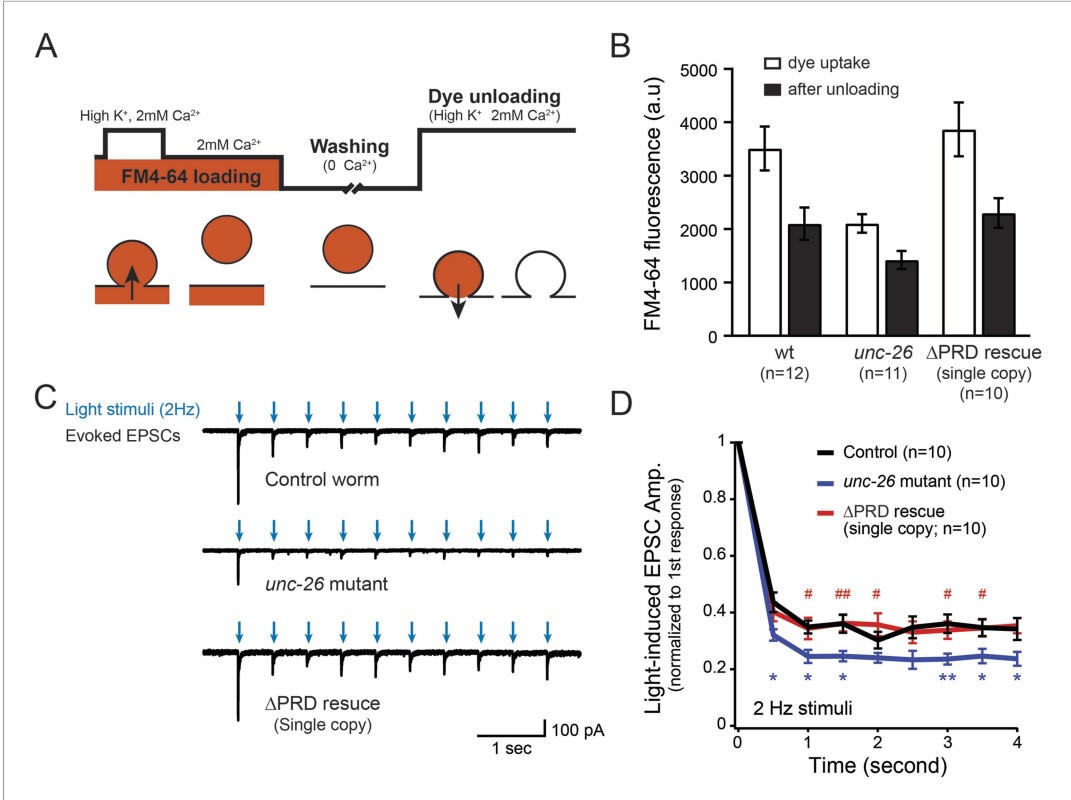

**Figure 2**. Synaptojanin UNC-26ΔPRD recovers the recycling vesicle pool and sustains synaptic transmission upon repetitive stimuli. (**A**) A schematic diagram is shown to illustrate the FM4-64 loading and unloading procedure. Experimental details are discussed in the 'Materials and methods' section. (**B**) FM4-64 loading and unloading at the head ganglion were compared for wt (n = 12), *unc-26* mutant (n = 11), and rescued worms with a single-copy transgene encoding GFP::UNC-26ΔPRD (n = 10). The expression of GFP::UNC-26ΔPRD significantly rescued both dye uptake and unloading (p < 0.01 and p < 0.01, respectively; compared to *unc-26 mutants*). (**C**) Acetylcholine currents were evoked by 2-Hz light pulses in worms carrying *Punc-17::ChR2::mCherry*. Representative traces of light-evoked EPSCs during repeated stimulation are shown for the indicated genotypes. (**D**) Mean values of currents normalized relative to the first EPSC were significantly reduced in *unc-26* mutant. The expression of GFP::UNC-26ΔPRD in *unc-26* mutants restores the amplitude of subsequent currents, suggesting that the UNC-26ΔPRD is functional to support synaptic transmission upon repeated stimuli. The number of worms analyzed for each genotype is indicated in the graph. *, p < 0.05 and **, p < 0.01 when compared to wt controls. #, p < 0.05 and ## p < 0.01 when compared to *unc-26* mutants. Error bars indicate SEM.

To further test the functional importance of the endophilin SH3, synaptojanin PRD interactions, we studied double-mutant worms that lack both *endophilin unc-57* and *synaptojanin unc-26*. Consistent with previous findings, synaptic defects in the *unc-57*; *unc-26* double-mutant worms were similar to *unc-57* and *unc-26* single mutants (*Schuske et al., 2003*) (*Figure 3* and *Table 1*), confirming that these genes function in the same genetic pathway. While the SH3-PRD scaffolding model predicts that SH3 and PRD are essential, we found that co-expression of single copies of mutant UNC-57 lacking SH3 (UNC-57ΔSH3::mCherry) and mutant UNC-26 lacking PRD (UNC-26ΔPRD::GFP) restores synaptic activities in *unc-57*; *unc-26* double mutants (*Figure 3C–F* and *Table 1*). Indeed, electron microscopy analyses show that the number of SVs was nearly normal in these animals (*Figure 4A*). Using quantitative Western blots, we found that mutant UNC-26ΔPRD and mutant UNC-57ΔSH3 were expressed at ~32% and ~75% of endogenous levels of UNC-26 and UNC-57, respectively, suggesting that the rescue activity of these transgenes was not due to compensatory artifacts of overexpression (*Figure 3—figure supplement 1*).

Finally, we utilized synaptopHluorin (SpH) to measure changes in surface synaptobrevin (*Dittman and Kaplan, 2006*; *Bai et al., 2010*). In SVs, SpH fluorescence is quenched by the acidic pH of the

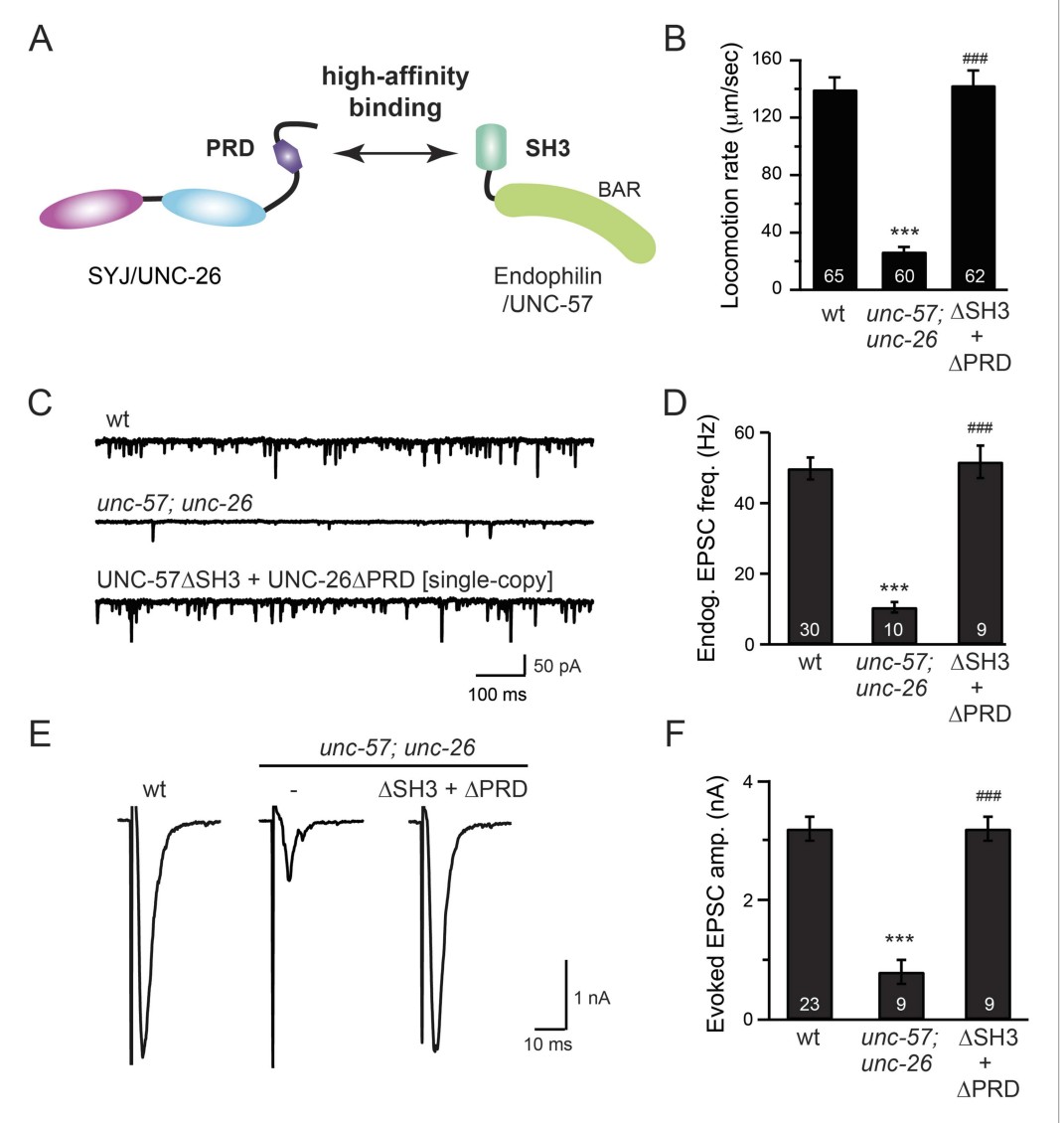

**Figure 3**. The SH3-PRD interaction is dispensable for synaptic activity. (**A**) A schematic drawing showing interactions between synaptojanin (UNC-26) PRD and endophilin (UNC-57) SH3. Single-copy transgenes encoding UNC-26ΔPRD::GFP and UNC-57ΔSH3::mCherry were co-expressed in *unc-57; unc-26* double mutants. Pan-neuronal promoters *Prab-3* and *Psnb-1* were used to drive expression of UNC-26ΔPRD::GFP and UNC-57ΔSH3::mCherry, respectively. Summary data for locomotion rate are shown in (**B**). Representative traces and summary data for endogenous EPSC rates (**C–D**) and for evoked EPSC amplitude (**E–F**) are shown for the indicated genotypes. The number of worms analyzed for each genotype is indicated in the bar graphs. ***, $p < 0.0001$ when compared to wt controls. ###, $p < 0.0001$ when compared to *unc-57; unc-26* double mutants. Error bars indicate SEM.

The following figure supplement is available for figure 3:

**Figure supplement 1**. UNC-57 and UNC-26 are not overexpressed in transgenic animals.

vesicle lumen. Following SV exocytosis, SpH fluorescence on the plasma membrane is dequenched (***Dittman and Ryan, 2009***). The *unc-57; unc-26* double mutants had a 63% increase in SpH axon fluorescence compared to control animals, consistent with a defect in recycling SV proteins from plasma membranes. Co-expression of UNC-57ΔSH3 and UNC-26ΔPRD fully rescued the SpH defects (***Figure 4B–D***), demonstrating that UNC-26ΔPRD and UNC-57ΔSH3 are functional to support SV endocytosis. Overall, these data demonstrate that endophilin and synaptojanin can support synaptic

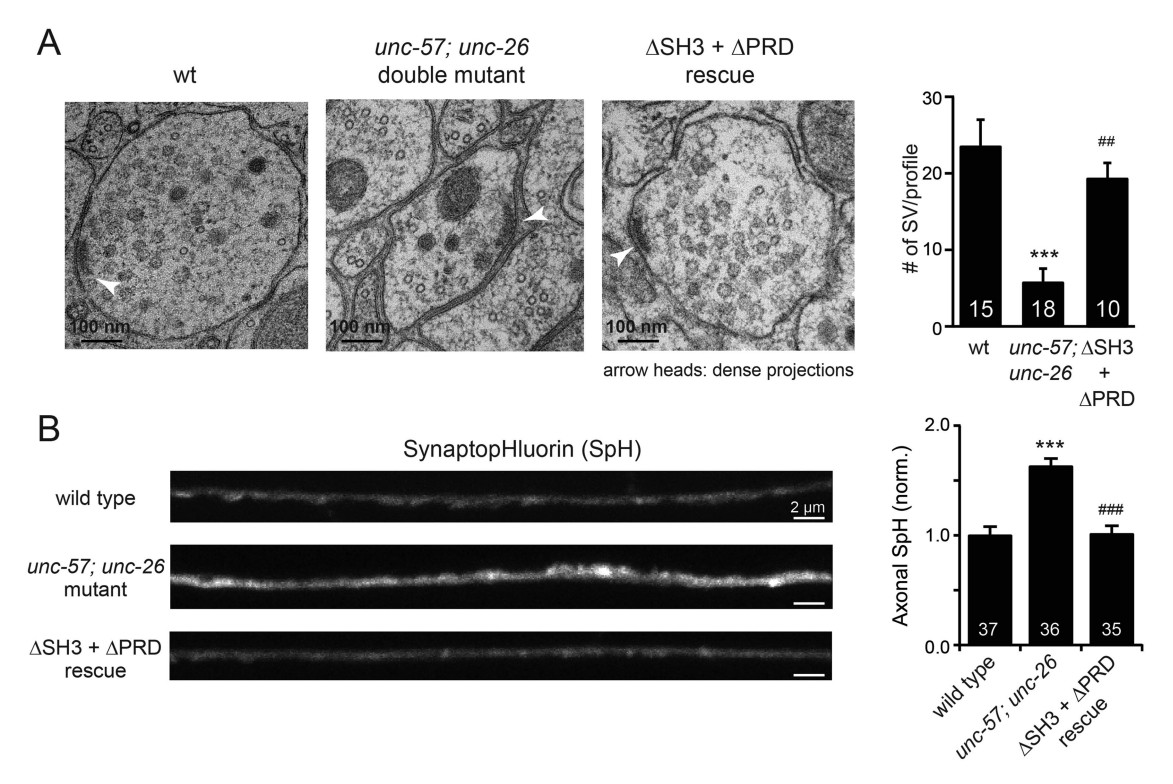

**Figure 4**. UNC-26ΔPRD and UNC-57ΔSH3 restore the number of SVs and recover synaptopHluorin retrieval in *unc-57; unc-26* double mutants. (**A**) Electron microscopy images of neuromuscular junctions were collected from the ventral nerve cords of adult hermaphrodites. Synaptic profiles of 15 synapses of the wt, 18 synapses of the *unc-57; unc-26* double mutants, and 10 synapses of the single-copy transgenic UNC-26ΔPRD::GFP; UNC-57ΔSH3::mCherry animals were analyzed. Arrowheads indicate dense projections. Synaptic vesicle (SV) number was counted in a blind manner. ***, p < 0.0001 when compared to wt controls. ###, p < 0.0001 and ##, p < 0.001 when compared to *unc-57; unc-26* double mutants. Scale bar: 100 nm. Error bars indicate SEM. (**B**) Representative images (*left*) and summary data (*right*) for axonal synaptopHluorin (SpH) fluorescence in the dorsal nerve cord are shown for the indicated genotypes. Rescue experiments are done using extrachromosomal arrays carrying *Psnb-1::unc-26ΔPRD* and *Prab-3::unc-57ΔSH3* (without any fluorescent tags). The number of worms analyzed for each genotype is indicated. ***, p < 0.0001 compared to wt controls. ###, p < 0.0001 when compared to *unc-57; unc-26* mutants. Scale bar: 2 μm. Error bars indicate SEM.

activity even in the absence of the SH3-PRD interaction. Therefore, additional uncharacterized mechanisms must exist to support synaptojanin function at synapses.

## Distinct roles of two phosphoinositide phosphatase domains of synaptojanin

We next investigated whether synaptojanin's unique configuration of tandem phosphoinositide phosphatase domains (*Figure 1A*) mediates the cooperation between synaptojanin and endophilin at synapses. The N-terminal Sac1 domain degrades phosphoinositides by hydrolyzing the 3- and 4-position phosphates (*Guo et al., 1999*), whereas the central 5-phosphatase domain converts PI(4,5)P2 into PI(4)P by removing the 5-position phosphate from the inositol ring (*Cremona and De Camilli, 2001*; *Chang-Ileto et al., 2011*). We found that inactivation of 5-phosphatase (D716A mutation) (*Whisstock et al., 2002*) completely abolished UNC-26 rescuing ability in restoring EPSC levels and locomotion (*Figure 5* and data not shown), indicating that the enzymatic activity of 5-phosphatase is required. By contrast, mutations (C378S,D380N) (*Guo et al., 1999*; *Hughes et al., 2000*) that inactivate Sac1 had little impact on UNC-26 activity, independent of the presence of the PRD domain (*Figure 5*, *Figure 5—figure supplement 1*, and *Table 1*). These data are consistent with previous reports showing that the mouse synaptojanin with inactivated Sac1 supports SV endocytosis in response to persistent activity (*Mani et al., 2007*), and that human patients with synaptojanin Sac1 mutations show no severe symptoms until reaching 20–40 years of age (*Krebs et al., 2013*). Together,

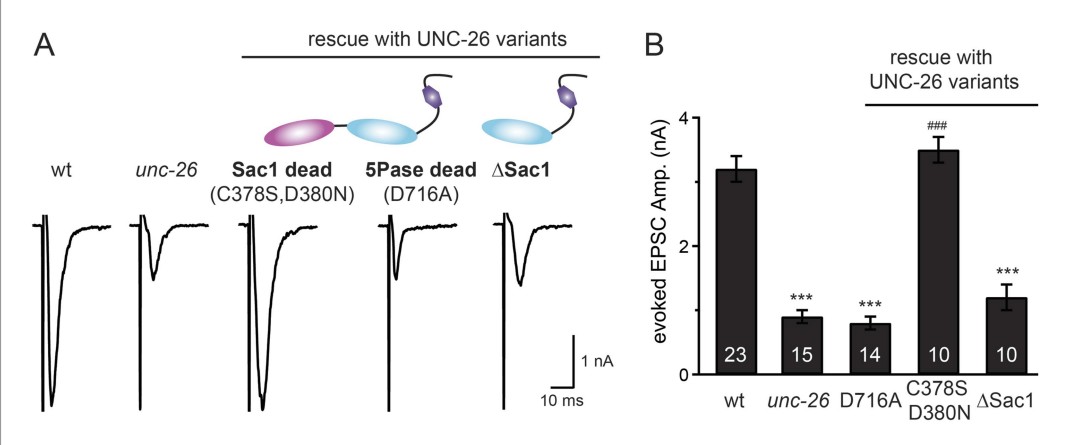

**Figure 5**. Synaptojanin phosphatase domains have distinct functions. The 5-Pase domain hydrolyzes phosphoinositides, while Sac1 plays an essential but non-catalytic role at synapses. Evoked EPSCs from wt, *unc-26*(*s1710*) mutant, and indicated transgenic strains were compared. Transgenes were GFP-tagged UNC-26 variants, including Sac1 dead (C378S, D380N; full-length UNC-26), 5-Pase dead (D716A, full-length UNC-26), and ΔSac1 (residues 494–1113). Transgenes were driven by *Prab-3*. Representative traces (**A**) and summary data (**B**) for evoked EPSC amplitudes are shown for the indicated genotypes. ***, $p < 0.0001$ when compared to wt controls. ###, $p < 0.0001$ when compared to *unc-26* mutants. The number of worms analyzed for each genotype is indicated in the bar graphs. Error bars indicate SEM.

The following figure supplement is available for figure 5:

**Figure supplement 1**. Sac1-inactivated synaptojanin supports synaptic transmission in a PRD independent manner.

these findings indicate that synaptojanin is able to support synaptic transmission, largely independent of its Sac1 phosphatase activity.

## Sac1 is an essential domain for synaptojanin activity

To ask whether the entire Sac1 domain plays any role in synaptojanin function, we generated a truncated UNC-26 that lacks the Sac1 domain (UNC-26ΔSac1, lacking residues 1–493). Surprisingly, we found that removal of the Sac1 domain severely disrupted the rescuing activity of UNC-26 (*Figure 5*), suggesting that the physical presence of Sac1 is required. Although isolated Sac1 and 5-phophatase fold correctly (*Tsujishita et al., 2001*; *Manford et al., 2010*), it remains possible that Sac1 deletion may perturb the folding of UNC-26. To address this issue in vivo, we used an intein-mediated protein ligation method to reconnect Sac1 to UNC-26 post-translationally (*Figure 6A*). We reasoned that if Sac1 truncation causes protein misfolding, UNC-26 would remain inactive after reconnecting with Sac1. However, if truncated UNC-26 fragments retain correct folding structure, protein ligation should lead to active full-length UNC-26.

We fused UNC-26 fragments with split DnaE intein from *Nostoc punctiforme* (*Npu*DnaE) (*Figure 6A*), as this intein system has been shown to be active in *C. elegans* (*Wong et al., 2012*). Co-expression of Sac1::Int[N] and Int[C]::UNC-26ΔSac1 significantly rescued the synaptic defects in *unc-26* mutants (*Figure 6B* and *Table 1*), suggesting that UNC-26 became functional upon post-translational ligation of Sac1. By contrast, transgenic worms that only express either Sac1::Int[N] or Int[C]::UNC-26ΔSac1 did not show functional improvements (*Figure 6—figure supplement 1* and *Table 1*). These data suggest that the two phosphatase domains of synaptojanin have distinct roles: the 5-phosphatase domain hydrolyzes phosphoinositides, while Sac1 plays a non-enzymatic role at synapses. Importantly, we found that Sac1 needs to be physically linked to UNC-26 to support synaptic transmission, as co-expression of UNC-26 fragments without the split inteins did not significantly rescue synaptic defects (*Figure 6B*).

## Sac1 targets synaptojanin to synapses

To gain insights into the non-enzymatic function of Sac1, we investigated the possibility that Sac1 guides 5-phosphatase for synaptic localization. We quantified synaptic abundance of GFP-tagged

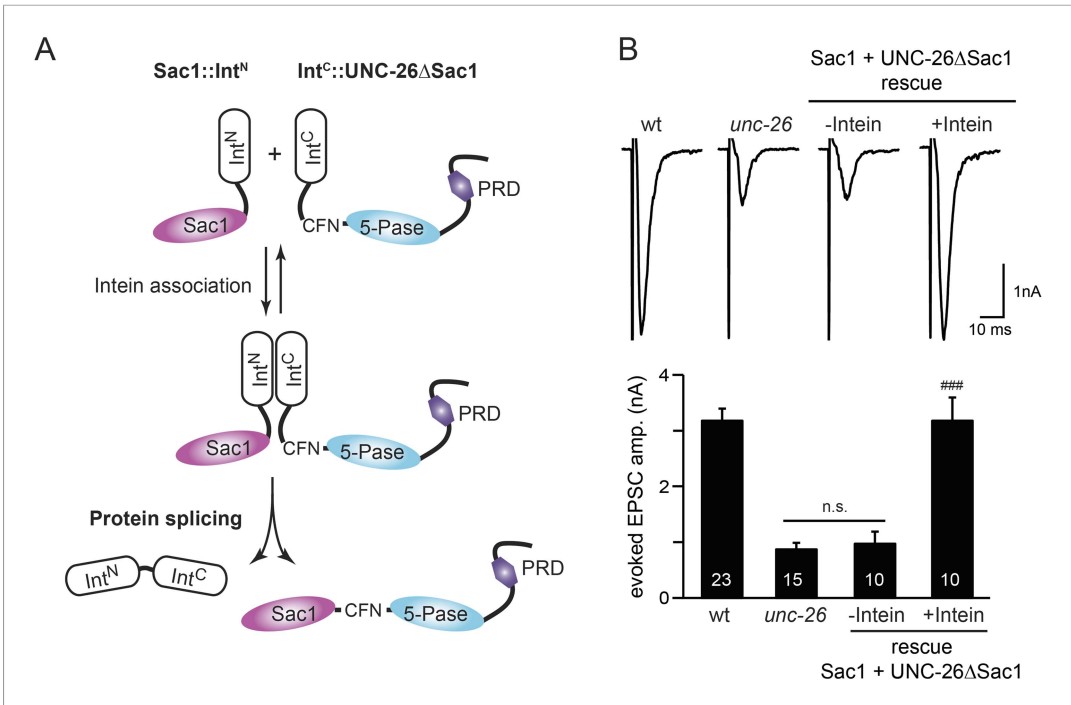

**Figure 6**. Sac1 must be physically linked to UNC-26 5-phosphatase to support synaptic transmission. Split-intein mediated ligation was used to post-translationally reconnect Sac1 to the remainder of the UNC-26 synaptojanin protein (**A**). The Sac1 domain (1–493) was linked to the N-terminal half of *Npu*DnaE to generate Sac1::Int$^N$. The C-terminal half of *Npu*DnaE was fused with the N-terminus of the UNC-26ΔSac1 fragment. Three extra residues (CFN) remain in the ligated product. Representative traces and summary data of evoked EPSCs are shown in (**B**). Co-expression of Sac1::Int$^N$ and Int$^C$::UNC-26ΔSac1 significantly rescued the synaptic defects in *unc-26* mutant worms. $^{###}$, p < 0.0001 when compared to *unc-26* mutants. The number of worms analyzed for each genotype is indicated in the bar graphs. Error bars indicate SEM.

The following figure supplement is available for figure 6:

**Figure supplement 1**. Transgenic worms that only express either Sac1::Int$^N$ or Int$^C$::UNC-26ΔSac1 did not show functional improvements.

---

UNC-26 variants. Full-length UNC-26 is enriched at synapses (synapse/axon ratio = 3.4 ± 0.1 fold, *Figure 7A–B*). However, removal of the Sac1 domain significantly reduced synaptic enrichment of UNC-26ΔSac1 (synapse/axon ratio = 2.4 ± 0.1 fold; *Figure 7A–B*), indicating that Sac1 has a critical role in retaining UNC-26 at synapses. Consistent with this idea, isolated Sac1 domains (both wt and the C378S,D380N mutant) are localized to synapses (*Figure 7—figure supplement 1*). It is likely that Sac1 and PRD act together to enhance synaptic distribution of synaptojanin, as deletion of both PRD and Sac1 domain further decreased synaptic enrichment (GFP::UNC-26ΔSac1ΔPRD synapse/axon ratio = 1.9 ± 0.1 fold; *Figure 7A–B*).

Interestingly, we found that UNC-26ΔPRD with inactivated 5-phosphatase (D716A), but not the version with wt phosphatase domains, exhibits significant levels of dominant-negative inhibition, presumably by competing with wt UNC-26. When D716A UNC-26ΔPRD was expressed in wt worms, the evoked EPSC amplitude was reduced by ~50% (*Figure 7C–D* and *Table 1*). The dominant negative effect was removed by elimination of the Sac1 domain (*Figure 7C–D*), indicating that the inactivated 5-phosphatase alone does not produce inhibitory activity. Together, these data argue that Sac1 plays a role in localizing synaptojanin to synapses.

## Bypassing Sac1 requirement by tethering with endophilin

Because our data suggest that Sac1 acts as a targeting domain rather than an enzyme, we speculated that it might be possible to bypass the Sac1 requirement by directly tethering the

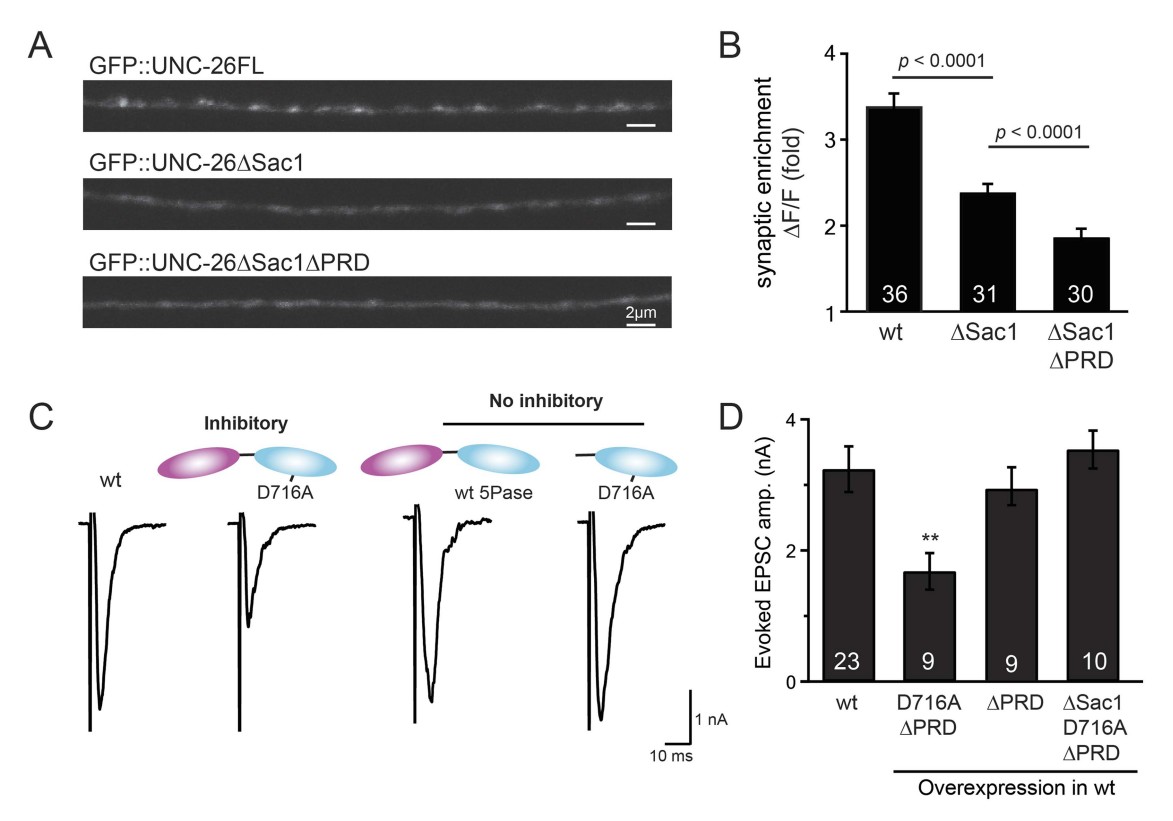

**Figure 7**. Sac1 is a synaptic targeting domain. (**A**–**B**) Removal of the Sac1 domain of synaptojanin perturbs synaptic targeting of synaptojanin. Representative images (**A**) showing various versions of GFP::UNC-26 distribution in the dorsal nerve cord. Synaptic enrichment of GFP::UNC-26 was quantified using $\Delta F/F = (F_{peak} - F_{axon})/F_{axon}$ and was compared for the indicated genotypes (**B**). Scale bar: 2 μm. (**C**–**D**) Sac1 is required for dominant negative inhibition. The D716A mutation that blocks 5-phosphatase activity was introduced into UNC-26ΔPRD and UNC-26ΔSac1ΔPRD. These UNC-26 variants were expressed in nervous system of wt worms. The stimulus-evoked EPSC amplitudes were significantly reduced in worms carrying UNC-26ΔPRD (D716A) mutant proteins. By contrast, animals expressing either UNC-26ΔPRD (with a functional 5Pase) or UNC-26ΔSac1ΔPRD (D716A) mutant proteins showed normal levels of synaptic activity. The number of worms analyzed for each genotype is indicated in the bar graphs. **, p < 0.001 when compared to wt controls. Error bars represent SEM.

The following figure supplement is available for figure 7:

**Figure supplement 1**. GFP-tagged Sac1 domains localize to synapses.

UNC-26ΔSac1 mutant with non-enzymatic proteins. We utilized three categories of proteins as candidate targeting tethers: (1) SV proteins SNB-1 (Synaptobrevin) and RAB-3; (2) lipid-binding domains that recognize specific phosphoinositides (*Lomasney et al., 1996*; *Várnai et al., 1999*; *Stahelin et al., 2007*); and (3) endocytic protein machinery adaptor AP2 subunits and accessory proteins (*Figure 8* and *Figure 8—figure supplement 1*). Among all proteins tested, UNC-57 endophilin was the only molecular tether that significantly restored UNC-26 activity in supporting endogenous EPSCs and evoked responses, no matter whether the PRD domain is present (*Figure 8A–E* and *Table 1*). The 5-phosphatase activity was still required for proper functioning of this chimeric UNC-57::UNC-26ΔSac1 protein, as the D716A mutation disrupted its ability to rescue synaptic defects (*Figure 8—figure supplement 2*). Overall, our findings indicate that the Sac1 domain has a non-enzymatic role in guiding synaptojanin 5-phosphatase, which can be replaced by endophilin.

## Endophilin BAR and synaptojanin Sac1 are functionally linked

We next investigated the molecular requirements for endophilin to functionally replace the Sac1 domain. Endophilin contains two domains: an N-terminal BAR domain that bends membranes, and

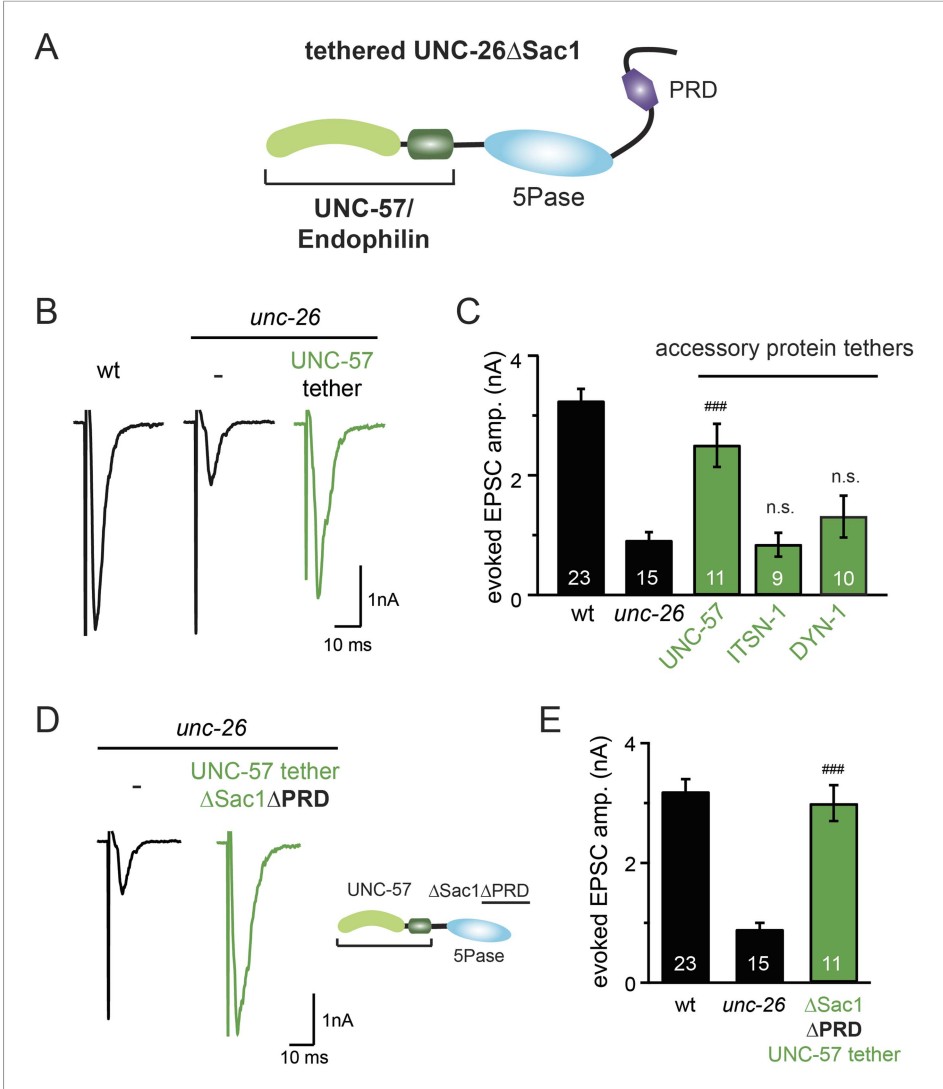

**Figure 8**. Endophilin functionally substitutes for the Sac1 domain. (**A**) A schematic drawing showing the chimeric UNC-57 endophilin::UNC-26ΔSac1 protein. Other endocytic accessory proteins including DYN-1 dynamin and ITSN-1 intersectin were tethered to UNC-26ΔSac1 using an identical strategy. Transgenes were expressed in all neurons using *Prab-3*. (**B–C**) Chimeric UNC-57 endophilin::UNC-26ΔSac1 proteins restore evoked EPSCs in *unc-26* mutant worms. Other tethers failed to rescue synaptojanin defects. Electrophysiological data in *Figure 8B–C* and *Figure 8—figure supplement 1* were collected blindly. (**D–E**) The PRD domain is not required for the endophilin tether to bypass the Sac1 requirement of synaptojanin. The number of worms analyzed for each genotype is indicated in the bar graphs. ###, $p < 0.0001$ when compared to *unc-26* mutants. 'n.s.' indicates $p > 0.05$ when compared to *unc-26* mutants. Error bars indicate SEM.
The following figure supplements are available for figure 8:

**Figure supplement 1**. Targeting UNC-26ΔSac1 to synaptic vesicles, phosphoinositides, and endocytic adaptor protein AP2 does not recover synaptojanin function.

**Figure supplement 2**. The endophilin tether does not bypass the requirement for a functional synaptojanin 5-phosphatase domain.

a C-terminal SH3 domain that interacts with PRD domains. We found that tethering UNC-26ΔSac1 to the endophilin BAR domain of either worm UNC-57 or rat endophilinA1 significantly restores synaptic transmission in *unc-26* mutants (*Figure 9A–B* and *Table 1*). In contrast, UNC-57 SH3 tethered

UNC-26ΔSac1 failed to rescue EPSC defects in *unc-26* mutants (data not shown), even though the SH3 domain enhanced the synaptic enrichment of UNC-26ΔSac1 (*Figure 9—figure supplement 1*). Together, these data indicate that endophilin BAR, rather than the SH3 domain, is the functional core for the Sac1 substitution. The functional difference between the BAR domain and the SH3 domain is likely due to their distinct binding partners and the potential for differential targeting to sub-synaptic regions.

Interestingly, we found that expression of a single copy of the *rEndoBAR::unc-26 5Pase* (*ΔSac1ΔPRD*) transgene significantly restored locomotion, endogenous EPSCs, double mutants (*Figure 9C–E* and *Table 1*), supporting the notion that synaptojanin UNC-26 and endophilin UNC-57 execute coincident action at synapses.

We next asked if BAR domains from other proteins could also functionally replace the Sac1 domain. When tethered to UNC-26ΔSac1, BAR domains from nadrin2 (*Galic et al., 2012*) and amphiphysin (*Peter et al., 2004*) slightly restored evoked EPSCs (*Figure 9—figure supplement 2*), indicating a trend of membrane-bending BAR domains to promote synaptojanin function.

While synaptojanin Sac1 and endophilin BAR do not bind each other in solution (*de Heuvel et al., 1997*; *Ringstad et al., 1997*) (*Figure 9—figure supplement 3*), they both bind membranes (*Guo et al., 1999*; *Farsad et al., 2001*). Therefore, we next asked whether chimeric UNC-26ΔSac1::BAR requires the BAR-membrane interactions for its function. Indeed, we found that the disruption of BAR-membrane interactions by either deleting the N-terminal amphipathic helix (ΔN) or by decreasing the positively charged residues (K76E,R78E) (*Gallop et al., 2006*) abolished rescue activity to restore synaptic transmission (*Figure 9A–B*). These data demonstrate that the membrane-binding activity of endophilin BAR is indispensible for Sac1 substitution. Together, these findings suggest that synaptojanin Sac1 and endophilin BAR are functionally coupled through membrane interactions to support SV recycling (*Figure 9F*).

## Discussion

Overall, our data suggest that Sac1, rather than PRD, plays a central role in synaptojanin function in vivo. This is unexpected because the current model proposes that the function of synaptojanin and endophilin requires high-affinity biochemical binding between endophilin SH3 and synaptojanin PRD. We further show that an endophilin anchor largely eliminates the requirement of the targeting role of Sac1, uncovering a new function of Sac1 in coupling synaptojanin and endophilin at synapses. We discuss the implications of these results below.

### The role of SH3-PRD interactions in SV endocytosis

Synaptojanin and endophilin are a classic example of the coordinated actions of endocytic proteins for rapid SV endocytosis (*Dittman and Ryan, 2009*; *Saheki and De Camilli, 2012*). While synaptojanin and endophilin have distinct biochemical properties, disruption of either protein leads to similar defects in SV recycling, indicating that synaptojanin and endophilin are functional partners in vivo (*Schuske et al., 2003*; *Verstreken et al., 2003*; *Van Epps et al., 2004*; *Dickman et al., 2005*). Currently, the molecular basis for such functional cooperation is attributed solely to the specific and high-affinity binding between the synaptojanin PRD domain and the endophilin SH3 domain (*de Heuvel et al., 1997*; *Ringstad et al., 1997*). Post-translational modifications such as phosphorylation may regulate SV endocytosis by controlling PRD-SH3 interactions (*Lee et al., 2004*; *Irie et al., 2005*). Although the current model suggests that the PRD-SH3 interactions are required, our data show that synaptojanin and endophilin remain active without the PRD and the SH3 domains, respectively. Similarly, mutant mouse synaptojanin that is defective in PRD-SH3 binding supports rapid endocytosis upon short stimuli (at 10 Hz) in mammalian neurons (*Mani et al., 2007*). These results show that the SH3-PRD interactions are not essential for synaptojanin activity in SV endocytosis.

Synaptojanin PRD and endophilin SH3 domains are positively selected and maintained during evolution, evident by the sequence conservation among animal species. This observation suggests that the SH3-PRD interactions are important for cellular activities; however, the precise role of these interactions is unclear. It is worth noting that our findings do not rule out a modulatory role of the SH3-PRD interactions at synapses. For example, in cultured hippocampal neurons, the SH3-PRD

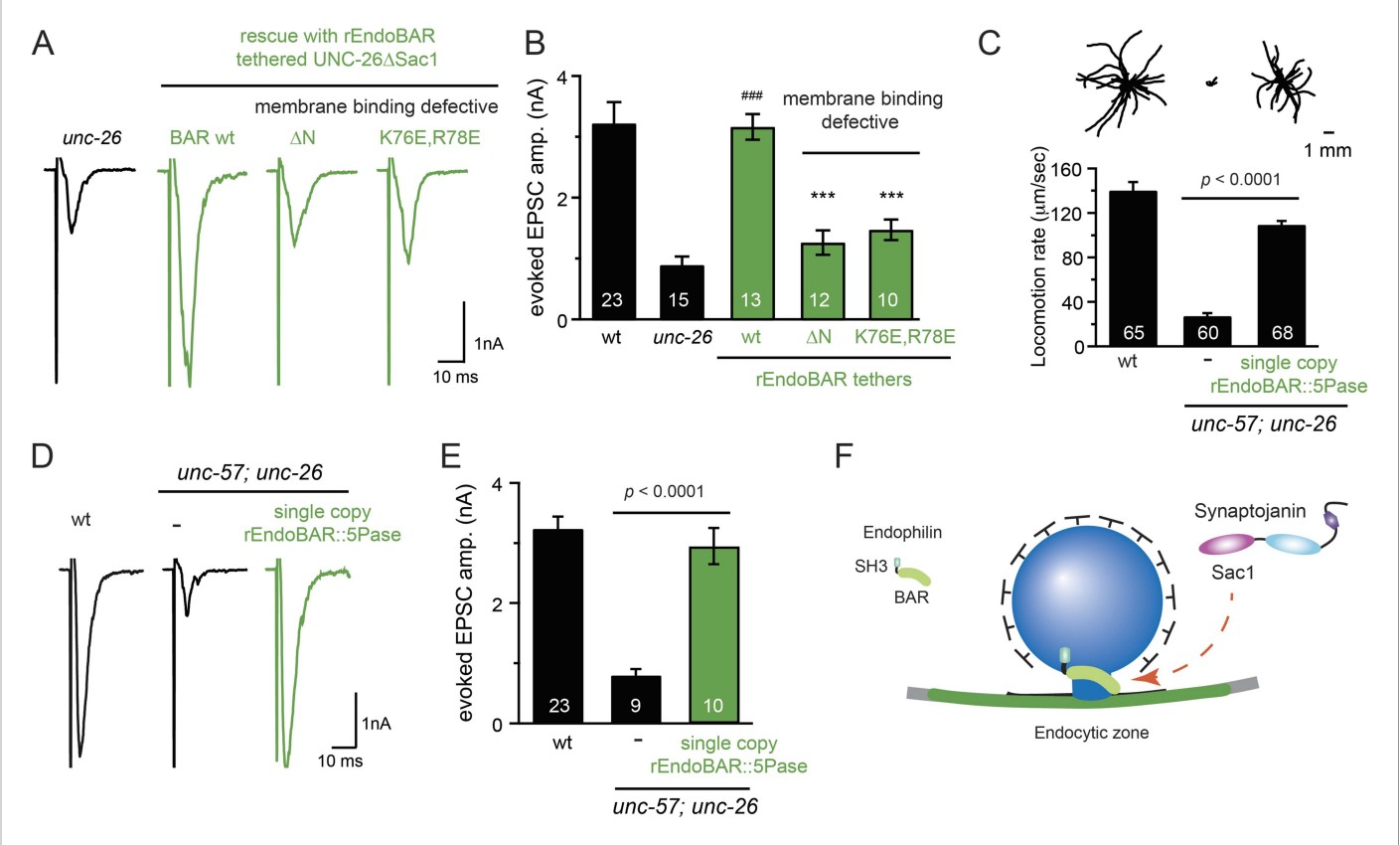

**Figure 9**. Endophilin BAR domain and its membrane interactions are required for bypassing Sac1. (**A–B**) Endophilin BAR is sufficient to bypass the Sac1 requirement. UNC-26ΔSac1 was tethered to worm and rat endophilinA1 BAR (rEndoBAR), respectively. Mutations that disrupt BAR-membrane interactions abolished the rescue activity of the chimeric UNC-26ΔSac1. ***, p < 0.0001 when compared to transgenic *unc-26* mutants carrying rEndoBAR wt::UNC-26ΔSac1. (**C–E**) Expression of a single-copy transgene encoding the rEndoBAR::UNC-26 5Pase chimera significantly restores locomotion and evoked EPSCs in *unc-57; unc-26* double mutants. Representative traces (*upper*) and summary data (*lower*) for locomotion (**C**), and evoked EPSCs (**D–E**) are shown for the indicated genotypes. ###, p < 0.0001 when compared to *unc-26* mutants. (**F**) A schematic diagram showing that the synaptojanin Sac1 domain and the endophilin BAR domain cooperate to promote SV endocytosis.

The following figure supplements are available for figure 9:

**Figure supplement 1**. UNC-57 SH3 domain enhances synaptic enrichment of UNC-26ΔSac1.

**Figure supplement 2**. Specificity in BAR proteins for bypassing the Sac1 requirement.

**Figure supplement 3**. UNC-26 Sac1 does not bind UNC-57 in solution.

interactions enhance the fidelity and speed of SV endocytosis after intense stimulation (*Mani et al., 2007*). Thus, it is possible that the PRD-SH3 interaction facilitates co-localization of endophilin and synaptojanin, helping the Sac1-dependent mechanism to sustain membrane recycling during persistent activity. In addition, while the PRD domain alone is not sufficient to promote SV endocytosis, it may become required in some situations, for example, when the Sac1-membrane interactions are reduced. Furthermore, PRD and SH3 domains may function in other important cellular processes, contributing to their conservation. Indeed, photoreceptor neurons in the zebrafish synaptojanin mutant exhibit significant defects in endosomes and the Golgi apparatus (*George et al., 2014*), suggesting that synaptojanin is needed for membrane-trafficking events at other intracellular organelles. The SH3-PRD interactions may be important for targeting membrane organelles in the cell body, where synaptojanin is less abundant than at synapses.

## Synaptojanin requires the unique configuration of tandem phosphatase domains

Our results indicate that the tandem phosphatase domains, Sac1 and 5-phosphatase, are essential for synaptojanin activity in SV endocytosis. The configuration of linked phosphatase domains is a unique feature of synaptojanin, and is reflected in its name derived from *janus*, the God of two faces (*Majerus and York, 2009*). Interestingly, while both phosphatase domains have catalytic activities in vitro (*Cremona et al., 1999*; *Guo et al., 1999*), the enzymatic activity of the 5-phosphatase domain is the only one required for synaptojanin function at synapses. This is consistent with previous findings showing that the major phosphoinositide defect in synaptojanin knockout mice is the abnormal accumulation of PI(4,5)P2 (*Cremona et al., 1999*). However, the 5-phosphatase activity alone is not enough. Expression of the 5-phosphatase, with or without the PRD domain, fails to restore synaptic activity. Unexpectedly, an enzymatic-dead version of Sac1 restores the activity of the 5-phosphatase domain to support synaptic transmission. These data suggest that the Sac1 domain possesses a novel targeting activity, which the PRD domain does not have.

Phosphoinositide phosphatases often harbor multiple lipid-binding domains to detect coincident signals for restricted localization on membranes (*Carlton and Cullen, 2005*). Synaptojanin is a highly dynamic protein that is transiently recruited to endocytic intermediates. The timing of synaptojanin recruitment is likely to be critical because SV endocytosis is a rapid process that occurs on the time scale of seconds (*Balaji and Ryan, 2007*). Our findings suggest that the targeting activity of Sac1 allows synaptojanin to recognize endocytic intermediates. In agreement with this notion, tethering synaptojanin 5-phosphatase to endophilin bypasses the requirement of the Sac1 domain and revives synaptojanin activity, suggesting that synaptojanin 5-phosphatase functions at sites where endophilin resides. Therefore, we propose that the Sac1 domain acts together with the 5-phosphatase as coincident detectors for membranes enriched in PI(4,5)P2 and endophilin.

## A membrane connection between synaptojanin Sac1 and endophilin

How the synaptojanin Sac1 domain recognizes endophilin-membrane complexes is currently unknown. Biochemical studies have shown that the Sac1 domain does not directly bind endophilin in solution (*Figure 9—figure supplement 3*) (*de Heuvel et al., 1997*; *Ringstad et al., 1997*). Here, we speculate that membranes serve as the molecular connection to couple these proteins, as both the Sac1 domain and the endophilin BAR domain bind membranes (*Guo et al., 1999*; *Farsad et al., 2001*). The endophilin BAR domain induces defects in lipid packing (*Gallop et al., 2006*) and consequently increases the exposure of lipid head groups. One possibility is that the synaptojanin Sac1 domain recognizes the lipid-packing defects generated by endophilin BAR, which in turn stimulates neighboring 5-phosphatase to degrade exposed PI(4,5)P2 head groups (*Chang-Ileto et al., 2011*). Alternatively, it is also possible that membranes stimulate direct binding between the Sac1 domain and the endophilin BAR domain. Nonetheless, our results show that the Sac1 domain is a crucial targeting domain for synaptojanin function. We propose that the Sac1 domain allows synaptojanin to detect endocytic membranes on which endophilin resides.

## Materials and methods

### Strains and DNA constructs

Strain maintenance and genetic manipulations were performed as described (*Brenner, 1974*). All *C. elegans* strains were maintained at 20°C on agar nematode growth media (NGM) plates seeded with OP50 bacteria. The N2 strain (Bristol, England) was used as wt. Mutant *unc-26(s1710)* and *unc-57 (e406)* strains were obtained from the Caenorhabditis Genetics Center and were subsequently outcrossed 10× times to the N2 strain. The following strains were used in this study:

BJH188 *unc-57(e406); unc-26(s1710)*

BJH180 *unc-26(s1710); pekSi8 [Prab-3::unc-26::gfp, cb-unc-119(+)]*

BJH88 *unc-26(s1710); pekSi7 [Prab-3::unc-26ΔPRD::gfp, cb-unc-119(+)]*

BJH40 *unc-26(s1710); pekEx15 [Psnb-1::mSyj1ΔPRD]*

BJH298 *unc-57(e406); pekSi19 [Psnb-1::unc-57ΔSH3::mCherry, cb-unc-119(+)]; unc-26(s1710); pekSi7 [Prab-3::unc-26ΔPRD::gfp, cb-unc-119(+)]*

BJH52 *unc-26(s1710); pekEx27 [Prab-3::gfp::unc-26(D716A)]*

BJH55 *unc-26(s1710); pekEx30 [Prab-3::gfp::unc-26(C378S,D380N)]*
BJH312 *unc-26(s1710); pekEx66 [Prab-3::gfp::unc-26ΔPRD(C378S,D380N)]*
BJH43 *unc-26(s1710); pekEx18 [Prab-3::unc-26Sac1, Prab-3::unc-26ΔSac1]*
BJH46 *unc-26(s1710); pekEx21 [Prab-3::unc-26ΔSac1]*
BJH49 *unc-26(s1710); pekEx24 [Prab-3::unc-26Sac1::Int^N]*
BJH48 *unc-26(s1710); pekEx23 [Prab-3::Int^C::unc-26ΔSac1]*
BJH145 *unc-26(s1710); pekEx39 [Prab-3::unc-26Sac1::Int^N, Prab-3::Int^C::unc-26ΔSac1]*
KP5105 *NuIs269 [Punc-129::gfp::unc-26]*
BJH360 *pekEx80 [Punc-129::gfp::unc-26ΔSac1]*
BJH53 *pekEx28 [Punc-129::gfp::unc-26ΔSac1ΔPRD]*
BJH338 *pekEx92 [Prab-3::gfp::unc-26ΔPRD(D716A)]*
BJH344 *pekEx98 [Prab-3::gfp::unc-26ΔSac1ΔPRD(D716A)]*
BJH310 *unc-26(s1710); pekEx64 [Prab-3::plc∂ PH:unc-26ΔSac1]*
BJH313 *unc-26(s1710); pekEx67 [Prab-3::bem1 PX::unc-26ΔSac1]*
BJH314 *unc-26(s1710); pekEx68 [Prab-3::btk PH::unc-26ΔSac1]*
BJH317 *unc-26(s1710); pekEx71 [Prab-3::aps-2::unc-26ΔSac1]*
BJH319 *unc-26(s1710); pekEx73 [Prab-3::itsn-1::unc-26ΔSac1]*
BJH320 *unc-26(s1710); pekEx74 [Prab-3::dyn-1::unc-26ΔSac1]*
BJH321 *unc-26(s1710); pekEx75 [Prab-3::apm-2::unc-26ΔSac1]*
BJH322 *unc-26(s1710); pekEx76 [Prab-3::apa-2::unc-26ΔSac1]*
BJH330 *unc-26(s1710); pekEx84 [Prab-3::unc-57::unc-26ΔSac1]*
BJH332 *unc-26(s1710); pekEx86 [Prab-3::unc-57::unc-26ΔSac1(D716A)]*
BJH333 *unc-26(s1710); pekEx87 [Prab-3::apb-1::unc-26ΔSac1]*
BJH335 *unc-26(s1710); pekEx89 [Prab-3::unc-57BAR::unc-26ΔSac1]*
BJH336 *unc-26(s1710); pekEx90 [Prab-3::rEndoBAR::unc-26ΔSac1]*
BJH337 *unc-26(s1710); pekEx91 [Prab-3::mAmphBAR::unc-26ΔSac1]*
BJH340 *unc-26(s1710); pekEx94 [Prab-3::rEndoBAR (K76E,R78E)::unc-26ΔSac1]*
BJH341 *unc-26(s1710); pekEx95 [Prab-3::rEndoBARΔN::unc-26ΔSac1]*
BJH343 *unc-26(s1710); pekEx97 [Prab-3::unc-26ΔSac1::snb-1]*
BJH345 *unc-26(s1710); pekEx99 [Prab-3::mNadrin2BAR::unc-26ΔSac1]*
BJH347 *unc-26(s1710); pekEx101 [Prab-3::unc-57::unc-26ΔSac1ΔPRD]*
BJH348 *unc-26(s1710); pekEx102 [Prab-3::unc-26ΔSac1::rab-3]*
BJH396 *zxIs6 [Punc-17::ChR2(H134R)::YFP; lin-15+], acr-16(ok789)*
BJH397 *zxIs6, acr-16(ok789); unc-26(s1710)*
BJH398 *zxIs6, acr-16(ok789); unc-26(s1710); pekSi7 [Prab-3::unc-26ΔPRD::gfp, cb-unc-119(+)]*
BJH399 *pekEx122 [Prab-3::gfp::unc-26ΔPRD]*
BJH400 *pekEx123 [Punc-129::gfp::unc-26Sac1(C378S,D380N)]*
BJH401 *pekEx124 [Punc-129::gfp::unc-57SH3::unc-26ΔSac1]*
BJH403 *pekEx126 [Punc-129::gfp]*
BJH405 *unc-57(e406); unc-26(s1710); pekSi24 [Psnb-1::rEndoBAR::unc-26ΔSac1ΔPRD; cb-unc-119(+)]*
BJH406 *pekEx127 [Punc-129::gfp::unc-26Sac1]*
BJH402 *nuIs122 [Pacr-2::synaptopHluorin]*
BJH407 *unc-57(e406); unc-26(s1710); nuIs122; pekEx125 [Psnb-1::unc-26ΔPRD; Prab-3::unc-57ΔSH3]*
BJH408 *unc-57(e406); unc-26(s1710); nuIs122*

Psnb-1 and Prab-3 promoters were used for neuronal rescue experiments, and Punc-129 for imaging analyses. cDNAs of unc-26, unc-57, snb-1, rab-3, dyn-1, apa-2, apb-1, apm-2, aps-2, and itsn-1 were amplified from total mRNA extracted from wt worms. cDNAs encoding rat endophilin A1, mouse Nadrin2, and mouse amphiphysin were amplified from a cDNA library from Clontech (Mountain View, CA, USA).

## Transgenes and germ line transformation

Transgenic strains for rescue experiments were generated by microinjection of various plasmids (2 ng μl⁻¹) together with co-injection markers, including *Pmyo-2::his11::gfp* (BJP-B36, 2 ng μl⁻¹), *Pvha-6::gfp* (BJP-B197, 10 ng μl⁻¹), *Pmyo2::NLS-MaxFP Green* (KP-JB473, 2 ng μl⁻¹), and *Pttx-3::DsRed* (KP-JB761,

50 ng µl$^{-1}$). For dominate-negative inhibition experiments, plasmids (BJP-M13, BJP-M82, and BJP-M185) were injected at ~60 ng µl$^{-1}$. For imaging experiments, variants of *Punc-129::gfp::unc-26* were injected at 15 ng µl$^{-1}$, unless specified. Blank vector pBluescript was used as an injection filler to bring final DNA concentration to 100 ng µl$^{-1}$. Integrated transgenes were obtained by UV irradiation of strains carrying extrachromosomal arrays. Transgenic worms were outcrossed at least 10 times.

## Single copy insertion of transgenes

Mos1-mediated single-copy transgene insertion methods were used to generate transgenic animals carrying single-copy transgenes (Frøkjaer-Jensen et al., 2012). The Mos1 target sites used in this study are ttTi5605 (chromosome II, for *unc-57* transgenes) and ttTi14024 (chromosome X, for *unc-26* transgenes). The following constructs were used to generate single-copy transgenes: BJP-B178 [*Prab-3::unc-26::gfp* for ttTi14024 (X)], BJP-B179 [*Prab-3::unc-26ΔPRD::gfp* for ttTi14024 (X)], BJP-B384 [*Psnb-1::unc-57ΔSH3::mcherry* for ttTi5605 (II)], and BJP-M208 [*Prab-3::rEndoBAR::unc-26 5Pase* for ttTi5605 (II)]. Transgenic worms carrying single copy insertion of transgenes were outcrossed at least 4 times.

## Worm tracking and analysis

Worm movement on 10 cm agar plates without bacterial lawn was recorded for 30 s. Young adults (reared at 20°C) were transferred to room temperature 1 hr prior to behavior tests. Videos of individual animals were captured on a CCD camera (MU130, AmScope, Irvine, CA) mounted on a stereomicroscope using 0.8× magnification. The center of mass was determined for each animal on each video frame using open-source object tracking scripts developed by Jesper S Pedersen (http://www.phage.dk/plugins/wrmtrck.html) in ImageJ (NIH, Bethesda, MD). Average speed was determined for each animal. Statistical analysis was performed using Igor Pro 6 (Wavemetrics, Lake Oswego, OR). Average values and standard error of the mean (SEM) were reported. p values were generated using one-way ANOVA followed by Dunnett's test.

## Electrophysiology

Young adult worms were immobilized on Sylgard-coated coverslips with cyanoacrylate glue (Histoacryl Blue, Aesculap, Center Valley, PA). Animals were dissected in extracellular solution via a dorsolateral incision. Gonads and intestines were removed to reveal the underlying ventral nerve cord and body-wall-muscle quadrants as previously described (Richmond et al., 1999; Bai et al., 2010). The worm prep was mounted onto a fixed stage upright microscope (BX51WI, Olympus, Japan) equipped with a 60× water-immersion objective lens.

Whole-cell patch clamp recordings were carried out at 20°C. A body wall muscle cell was voltage clamped at −60 mV to record postsynaptic currents. Evoked EPSC responses were induced by applying a 0.4 ms, 30 µA pulse, generated by a stimulus isolator (A365, WPI, Sarasota, FL), through a borosilicate pipette (~2 MΩ) placed in close apposition to the ventral nerve cord. Series resistance was compensated to 70% for the evoked EPSC recording. The currents were amplified using EPC-10 (HEKA, Germany). The signals were sampled at 10 kHz using Patchmaster (HEKA) following low-pass filtering at 2 kHz. Patch pipettes (2–5 MΩ) were pulled using borosilicate glass and were fire polished. Extracellular solution contains (in mM) 150 NaCl, 5 KCl, 1 CaCl2, 5 MgCl2, 10 glucose, and 10 HEPES and was titrated to pH 7.3 with NaOH, 330 mOsm with sucrose. Internal solution contains 135 CH3O3SCs, 5 CsCl, 5 MgCl2, 5 EGTA, 0.25 CaCl2, 10 HEPES, and 5 Na2ATP and was adjusted to pH 7.2 using CsOH. All chemicals were purchased from Sigma (St. Louis, MO).

Electrophysiological data were analyzed with open-source scripts developed by Eugene Mosharov (http://sulzerlab.org/Quanta_Analysis_8_20.ipf; Mosharov and Sulzer, 2005) in Igor Pro 6 (Wavemetrics). Average values and SEM were reported. Statistical analysis was performed using Igor Pro 6. p values were generated using one-way ANOVA followed by Dunnett's test. A p-value < 0.05 was considered to be significant.

## Retinal feeding

NGM plates (60 mm) were seeded with 250 µl of OP50 bacteria and 4 µl of 100 mM all-trans retinal (Sigma, St. Louis, MO). Seeded retinal plates were kept in the dark at 4°C and were used within 7 days. Channelrhodopsin-2 transgenic worms (L4 hermaphrodites) were transferred from regular plates to retinal plates in the dark at room temperature and then grown for an additional 16 hr before electrophysiological experiments.

## Light stimulation

A TILL Oligochrome light source (Till Photonics, Germany) was controlled by TTL signals from a HEKA EPC-10/2 amplifier. Blue light (460–500 nm) through a GFP filter set (49012, Olympus) was used to excite channelrhodopsin-2. The light intensity output from the 60× objective (1.0 NA) was 12 mW/mm$^2$, quantified by an XR2100 power meter (X-Cite, Canada). Light pulses (8 ms duration, 2 Hz) were used to evoke post-synaptic currents at neuromuscular junctions.

## Transmission electron microscope

Approximately, 5 adult hermaphrodites were loaded at room temperature into a 100 μm specimen chamber containing space-filling bacteria and M9 buffer. These worms were frozen instantaneously at ∼ −180°C in either a Leica EM PACT2 (Leica, Germany) or a BAL-TEC HPM010 (Bal-Tec, Liechtenstein) system. The frozen worms were fixed in a Leica EM AFS2 machine using 1% osmium in 0.1% UA in acetone fixative and then embedded in Eponate 12 from Ted Pella, Inc (Redding, CA). Serial sections were cut at a thickness of 40 nm, collected on pioloform covered slotted grids (notchnum 1 × 2 mm oval) from Ted Pella, Inc., and counterstained in 6% aqueous uranyl acetate for 1.5 hr, followed by Reynolds lead citrate for 7 min. Images were obtained on a JEOL JEM 1400 transmission electron microscope (JEOL, Japan) operating at 120 KV. Micrographs were collected using the Gatan Ultrascan 1000XP, 2k × 2k high-resolution camera (Pleasanton, CA). Synapse profiles were used to count the number of synaptic vesicles (∼30 nm in diameter). Each profile represents a single section that passes through the dense projection. Vesicle counting was performed blindly. p values were generated using one-way ANOVA followed by Dunnett's test.

## In vivo microscopy and image analysis

Animals were immobilized with 2,3-Butanedione monoxamine (30 mg ml$^{-1}$; Sigma–Aldrich) and were mounted on 2% agarose pads for imaging. Fluorescence images were collected on an inverted Olympus FV-1000 confocal microscope with an Olympus PlanApo 60× Oil 1.4 NA objective at 5× zoom. GFP was excited using a 488 nm argon laser (0.5% laser power). Images of fluorescent slides (Chroma Technology Group, Rockingham, VT) were captured during each imaging session to provide a fluorescence standard for comparing fluorescence intensities between animals. Line scans were analyzed with custom-written scripts developed by Jeremy Dittman (Weill Cornell Medical College; *Dittman and Kaplan, 2006*) in Igor Pro (Wavemetrics, OR). Background signal was subtracted before analysis. 'Synaptic enrichment' (% $\Delta F/F$) is defined as $(F_{peak} - F_{axon})/F_{axon}$. All the values reported in the figures are mean ± SEM.

## FM4-64 loading and unloading

Young adult worms were immobilized in standard medium (150 mM NaCl, 5 mM KCl, 2 mM CaCl2, 4 mM MgCl2, 10 mM glucose, and 10 mM HEPES [pH 7.3]) on Sylgard-coated coverslips with cyanoacrylate glue (Histoacryl Blue, Aesculap). The head neuron ganglion was exposed by a small incision using a sharp needle. After the ganglion was exposed, a second incision was made at the middle section of worm body to release internal pressure. Dissected worms were gently rinsed with standard medium. To stimulate FM4-64 (Invitrogen, Carlsbad, CA) loading, dissected worms were incubated with high-potassium buffer (85 mM KCl, 70 mM NaCl, 2 mM CaCl2, 4 mM MgCl2, 10 mM glucose, and 10 mM HEPES [pH 7.3]) in the presence of 10 μM FM4-64 dye for 1 min. Stimulated preparations were then incubated with standard medium containing 10 μM FM4-64 dye for 2 min to allow for vesicle recycling to proceed. To remove surface-bound dye, dissected worms were gently washed in a Ca$^{2+}$-free low-K$^+$ buffer (0.5 mM EGTA and 1 mM ADVASEP-7 [Sigma]) for 5 min. Dye unloading from releasable vesicles was carried out by incubation with high-potassium medium without FM4-64 dye for 5 min. Imaging was done on a Zeiss LSM 780 confocal microscope (Zeiss, Germany) with a 40×/0.8 objective. FM4-64 was excited with a 561-nm laser (1% laser power), and fluorescence emission was collected between 643 nm and 751 nm. A set of Z-stack images (9–12 sections, step size 1.71 μm) was obtained for each worm. Images were imported into ImageJ for data analysis (NIH). For all images in each Z-stack, an area of interest (AOI) was defined by drawing a circle of 45 μm in diameter around the neuron ganglion. Total fluorescence of each image was obtained by integrating fluorescence pixel intensity within the AOI. Background fluorescence was subtracted from all images. The fluorescence signal (arbitrary unit, a.u.) with the highest value from each stack was used for

comparison. Statistical analyses were performed using one-way ANOVA followed by Dunnett's test. All the values reported in the figures are mean ± SEM.

## Recombinant protein purification

All versions of recombinant UNC-57 and UNC-26 proteins were expressed in an *Escherichia coli* strain BL21(DE3) as fusion proteins. DNA fragments encoding UNC-57 full-length, UNC-57 BAR (residues 1–283), and UNC-57 SH3 (residues 284–379) were inserted into pGEX4T-1 using BamHI and NotI sites. Recombinant GST::UNC-57 variants were immobilized onto glutathione beads (Genscript, Piscataway, NJ). UNC-26 Sac1 (residues 1–493), 5-phosphatase (residues 494–986), and PRD (residues 987–1113) were fused to the C-terminus of the maltose-binding protein (MBP) using overlapping PCR. DNA fragments encoding MBP::UNC-26 variants were subcloned into PET28a using NdeI and XhoI sites to produce C-terminal His6-tagged fusion proteins. Recombinant MBP-UNC-26Sac1-his6 proteins were purified using Ni-NTA Agarose (Qiagen, Valencia, CA) and were eluted in HEPES buffer (50 mm HEPES, pH 7.4, 150 mM NaCl) plus 250 mM imidazole. For antibody development, DNA encoding UNC-57 BAR::mCherry was inserted into PET28 using BamHI and NotI sites, and DNA encoding SUMO::UNC-26 (residues 494–986) was inserted into PET28 using NcoI and NotI sites. Purification of GST- and His6-tagged proteins was performed essentially as previously described (*Bai et al., 2004*).

## Pull-down assays

GST-pull down assay was performed as previously described with modification (*Bai et al., 2004*). GST-tagged UNC-57 proteins (10 µg) were immobilized on glutathione beads (Genscript). Recombinant MBP::UNC-26::His6 fragments (2 µM) were then incubated with beads in a binding buffer composed of 20 mM HEPES, 150 mM NaCl, 1% Triton X-100, and 1 mM Dithiothreitol (DTT). After 2 hr, the beads were washed 3 times with binding buffer, and the sample was treated with SDS sample buffer, subjected to SDS-PAGE, and visualized by staining with Coomassie Brilliant Blue G-250.

## Monoclonal antibody production

Monoclonal antibodies to *C. elegans* UNC-57 BAR and UNC-26 5-phosphatase proteins were generated in the FHCRC Monoclonal Antibody Core Facility. Recombinant proteins (UNC-26 5-phosphatase [residues 467–986] and UNC-57ΔSH3 [residues 1–283]) were used as antigens. Mice (e.g., Swiss Webster, A/J, and C57BL/6) were immunized, and immune splenocytes were isolated from mice showing positive antisera titers (ELISA, enzyme-linked immunosorbent assay). Isolated splenocytes were electrofused (BTX, Harvard Apparatus, Holliston, MA) to the NS-1/FOX-NY myeloma cell line. Antibody secreting hybridomas were identified using standard ELISA screens, and monoclonal hybridomas were isolated by limiting dilution subcloning. Monoclonal antibodies to UNC-57 and UNC-26 were further screened and characterized by Western blot analysis of recombinant proteins and *C. elegans* detergent extracts.

## Worm lysis and Western blot

Worms were reared on 10–15 enriched peptone plates (15 cm) at 20°C. Adult animals were collected using the sucrose flotation method, frozen in liquid nitrogen, and homogenized for 20 s using a mini bead-beater16 (Biospec Product, Bartlesville, OK) in M9 buffer with 2 mM Ethylenediaminetetraacetic acid (EDTA), 1 mM DTT, and 1× protease inhibitor cocktail (Sigma). Worm samples were then cooled for 1 min on ice. The beat/cool cycle was repeated 6 times to completely lyse the worms. Triton-X100 (final 1%) was next added to extract total protein. After 15 min of incubation, crude protein extracts were centrifuged at 14,000 rpm in an Eppendorf centrifuge (5417C, Eppendorf AG, Germany) for 10 min at 4°C to remove the beads, worm debris, and unbroken worms. The protein concentration of worm extracts was determined by the Pierce bicinchoninic acid (BCA) assay (Thermo Scientific, Waltham, MA). Primary antibodies against UNC-57BAR and UNC-26 5-phophatase were used for Western blot. Immunoreactive bands were visualized using enhanced chemiluminescence and were quantified using a Bio-Rad ChemiDoc MP imaging system (Bio-Rad, Hercules, CA).

## Acknowledgements

We thank Drs Jonathan Cooper, Jeremy Dittman, Joshua Kaplan, Harmit Malik, Meng-Chao Yao, Dan Gottschling, and members of the Bai lab for comments. Some strains were provided by the

Caenorhabditis Genetics Center, which is funded by NIH Office of Research Infrastructure Programs (P40 OD010440). We thank the Kaplan Lab for strains and reagents. We thank Kent McDonald and FHCRC shared resources for technical help in generating transgenic worms, antibody development, and electron microscopy.

## Additional information

### Funding

| Funder | Grant reference | Author |
| --- | --- | --- |
| National Institutes of Health (NIH) | R00-MH085039 | Jihong Bai |
| National Institutes of Health (NIH) | R01-NS085214 | Jihong Bai |

The funder had no role in study design, data collection and interpretation, or the decision to submit the work for publication.

### Author contributions

YD, Conception and design, Acquisition of data, Analysis and interpretation of data, Drafting or revising the article; YG, YL, Acquisition of data, Contributed unpublished essential data or reagents; YL, Acquisition of data, Analysis and interpretation of data; JB, Conception and design, Drafting or revising the article, Contributed unpublished essential data or reagents

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
