## [Decision Letter]

Thank you for sending your work entitled “Synaptojanin cooperates in vivo with endophilin through an unexpected mechanism” for consideration at *eLife*. Your article has been favorably evaluated by Randy Schekman (Senior editor), a Reviewing editor, and three reviewers.

The following individuals responsible for the peer review of your submission have agreed to reveal their identity: Graeme Davis (Reviewing editor) and Erik Jorgensen (peer reviewer). A further reviewer remains anonymous.

The Reviewing editor and the reviewers discussed their comments before we reached this decision, and the Reviewing editor has assembled the following comments to help you prepare a revised submission.

Summary:

The manuscript by Dong et al., investigates the involvement of synaptojanin/UNC-26 and endophilin/UNC-57 in synaptic vesicle endocytosis at the NMJ in *C. elegans*. While these two proteins are well-established factors in the synaptic vesicle cycle (knockouts of either one severely impair synaptic transmission), how or whether they cooperate in this process is less well understood. There is published biochemical evidence for direct binding between endophilin and synaptojanin (SYNJ) via their SH3 and proline rich domains respectively, and this has led to a model where a direct interaction between these domains underlies their function in synaptic endocytosis and maintenance of the vesicle pool. This manuscript centers on the surprising result that the SYNJ PRD (and therefore its direct interaction with endophilin) is dispensible for its function in supporting normal synaptic transmission. Instead, SYNJ's Sac1 domain appears to play an important role (independent of its phosphatase activity) and is potentially required for enriching it at synapses. Overall, the experiments are solid, using a UNC-26 genetic deletion/transgene rescue approach with various mutants and truncations. The novel and non-catalytic role of the Sac1 domain and the surprising result that SYNJ's PRD is not essential for its role in synaptic transmission make this work conceptually interesting and challenges the existing model. The potential applicability of this mechanism to other forms of membrane trafficking make the work of broader interest beyond the synaptic vesicle trafficking field. While the role of the Sac1 domain remains mysterious, this domain is potentially interesting in light of recent evidence for a SYNJ Sac1 mutation (which impairs it's catalytic activity) in a family with early onset Parkinsonism.

Required modifications:

1) For several experiments, the authors show only evoked EPSCs to document the function of chimeric proteins. While not a strict requirement for publication, it would be nice to also show mEPSCs and locomotion data whenever possible. (The mEPSC data likely exist already.)

2) There are no direct measures of endocytosis presented in the manuscript. Two modifications would substantially strengthen the manuscript and make the train of logic more accessible to the general reader. First, it is important to show a direct measurement of endocytosis to establish that the defects in exocytosis can be ascribed to a defect in endocytosis. Second, the authors should be clear, in the text, that the defect in exocytosis is believed to be due to the cumulative effects of impaired endocytosis over time, resulting in diminished vesicle number and a diminished releasable vesicle pool. If there is another interpretation that can explain this association, this if fine, but should be explicitly stated. In dealing with this issue the text should also include reference to Ch'ng et al., PLoSG (2008) demonstrating that RIM puncta density is not impaired in the *unc-26* mutant, thus providing evidence that smaller EPSCs is not a consequence of fewer synapses.

3) Methods. Experimental procedures are not well described and will be hard for others to replicate the work. Please provide more details in each section.

4) N-values are not apparent throughout. How many synapses were analyzed? How many animals were analyzed? Were the experiments blinded?

Additional suggestions to improve the impact of this study:

1) Rescue of *unc-57 unc-26* double mutant by a single-copy of the BAR-5-phosphatase chimera would be elegant if the authors can provide it.

2) Is the isolated Sac1 domain (WT and catalytic mutant versions) synaptically localized? This would be a nice way of documenting its function as a localizing domain. In other words, is the Sac1 sufficient to localize GFP to synapses in the wild-type?

3) In Figure 5, the authors show that the D716A 5-Phosphatase mutant has dominant negative activity, and that this inhibitory activity requires the Sac1 domain. This construct lacks the PRD. Have the authors tested if a ∆PRD construct containing wild type phosphatase domains also has dominant negative activity? This control would help conclude that the D716A mutation is required for the inhibitory action of this construct.

4) In the subsection headed “Endophilin BAR and synaptojanin Sac1 are functionally linked”, the authors state that fusing Endophilin's SH3 domain to UNC-26∆Sac did not rescue EPSC defects. Did the SH3 domain restore synaptic localization of ∆Sac? This would suggest that proper localization of the 5-Phosphatase domain is not sufficient to explain UNC-26 function in endocytosis.

5) In the same section of the manuscript, the authors speculate as to why the SH3 and PRD domains are conserved despite the fact that they are not required for SV endocytosis. Another possibility the authors may want to suggest is that the PRD/SH3 interaction provides a backup mechanism for co-localizing Endophilin and SJN under conditions where Sac1 localization is disrupted. For example, if Sac1 binds PI3,4P lipids, its localization may be disrupted when the levels of these lipids are low. In this case, the SH3/PRD interaction may be required for endocytosis.

6) In Figure 2, what do the arrowheads indicate?

7) In Figure 7, panel C, it is unclear what the cyan domain indicates in the BAR domain containing protein.

8) In Figure 7—figure supplement 1, was rescue done with high copy or single copy transgenes?

---

## [Author Response]

*1) For several experiments, the authors show only evoked EPSCs to document the function of chimeric proteins. While not a strict requirement for publication, it would be nice to also show mEPSCs and locomotion data whenever possible. (The mEPSC data likely exist already*.*)*

We have added a table (Table 1) and Figure 9 to disclose all available data on locomotion, endogenous EPSCs, and evoked EPSCs.

*2) There are no direct measures of endocytosis presented in the manuscript. Two modifications would substantially strengthen the manuscript and make the train of logic more accessible to the general reader. First, it is important to show a direct measurement of endocytosis to establish that the defects in exocytosis can be ascribed to a defect in endocytosis. Second, the authors should be clear, in the text, that the defect in exocytosis is believed to be due to the cumulative effects of impaired endocytosis over time, resulting in diminished vesicle number and a diminished releasable vesicle pool. If there is another interpretation that can explain this association, this if fine, but should be explicitly stated. In dealing with this issue the text should also include reference to Ch'ng et al., PLoSG (2008) demonstrating that RIM puncta density is not impaired in the* unc-26 *mutant, thus providing evidence that smaller EPSCs is not a consequence of fewer synapses*.

Several changes have been made to address these concerns.

We now show results from two endocytosis assays. First, we found that expression of UNC-26∆PRD rescues the FM4-64 uptake in *unc-26* mutants (Figure 2). The following text was added to the manuscript:

“To assay for membrane recycling, we employed FM4-64, a fluorescent lipophilic dye that is internalized by endocytosis […] indicating that the recovery of vesicle recycling processes does not require UNC-26PRD.”

Second, we show that co-expression of endophilin∆SH3 and synaptoajanin∆PRD restores the surface amount of synaptopHluorin in the *unc-57; unc-26* double mutants (Figure 4). We added the following text to the manuscript:

“Finally, we utilized SynaptopHluorin (SpH) to measure changes in surface synaptobrevin […] demonstrating that UNC-26∆PRD and UNC-57∆SH3 are functional to support SV endocytosis.”

Third, to clarify the link between impaired endocytosis and diminished releasable vesicle pool, we have added the following text to the manuscript:

“These defects are consistent with previous reports showing reduced SV pools and a corresponding decrease in synaptic transmission due to the cumulative effects of impaired endocytosis over time.”

Fourth, to further strengthen the link between endocytosis and exocytosis, we have assayed for changes of synaptic transmission upon repetitive light stimuli, using worms carrying Channelrhodopsin-2 transgenes. We show that control synapses and synapses rescued by truncated UNC-26∆PRD sustain synaptic transmission to similar levels upon repetitive stimuli. We have added the following text to the paper:

*“*Cholinergic neurons of transgenic animals carrying *Punc-17::ChR2(H134R)::YFP* were activated by 2Hz photostimulation […] supporting the notion that truncated UNC-26 functions sufficiently to supply SVs during sustained activity.”

Finally, we have added the citation of Ch'ng et al., PLoSG (2008) to support the notion that smaller EPSCs is not a consequence of fewer synapses. It now reads:

“Because the density of active zone markers (e.g., RIM/UNC-10) remains unchanged in *unc-26* mutants (7), reduced EPSC frequency and amplitude cannot be explained by fewer synapses.*”*

*3) Methods. Experimental procedures are not well described and will be hard for others to replicate the work. Please provide more details in each section*.

We have expanded the Methods to provide more details.

*4) N-values are not apparent throughout. How many synapses were analyzed? How many animals were analyzed? Were the experiments blinded*?

All sample size values are now shown on the figures, as well as in the table and figure legends. Synaptic vesicle counting (Figure 4) and the initial screen for chimeric proteins (Figure 8 and Figure 8—figure supplement 1) were carried out in a blind manner.

*Additional suggestions to improve the impact of this study*:

*1) Rescue of* unc-57 unc-26 *double mutant by a single-copy of the BAR-5-phosphatase chimera would be elegant if the authors can provide it*.

We have generated transgenic worms that carry a single-copy of the *Psnb-1::rEndoBAR::UNC-26 5Pase* transgene. We found that rEndoBAR::UNC-26 5Pase chimera significantly rescues locomotion, endogenous EPSC frequency, and evoked EPSC amplitude in the *unc-57; unc-26* double mutants. Interestingly, the single-copy transgene appears to rescue locomotion and endogenous EPSCs to a better degree than the high-copy simple arrays reported in the previous version of the manuscript. This difference may be due to the mosaicism in transgenic worms carrying simple high-copy extrachromosomal arrays.

The text has been modified as follows: “we found that expression of a single copy of the *rEndoBAR::unc-26 5Pase (∆Sac1∆PRD)* transgene significantly restored locomotion, endogenous EPSCs, double mutants (Figure 9 and Table 1), supporting the notion that synaptojanin UNC-26 and endophilin UNC-57 execute coincident action at synapses.”

*2) Is the isolated Sac1 domain (WT and catalytic mutant versions) synaptically localized? This would be a nice way of documenting its function as a localizing domain. In other words, is the Sac1 sufficient to localize GFP to synapses in the wild-type*?

We show that isolated Sac1 domains (both wild type and the C378S,D380N mutant) are sufficient to localize GFP to synapses in wild type worms (Figure 7—figure supplement 1).

*3) In*
Figure 5*, the authors show that the D716A 5-Phosphatase mutant has dominant negative activity, and that this inhibitory activity requires the Sac1 domain. This construct lacks the PRD. Have the authors tested if a ∆PRD construct containing wild type phosphatase domains also has dominant negative activity? This control would help conclude that the D716A mutation is required for the inhibitory action of this construct*.

We show that the UNC-26∆PRD with wild type phosphatase domains does not exhibit dominant-negative inhibition (Figure 7).

*4) In the subsection headed “Endophilin BAR and synaptojanin Sac1 are functionally linked”, the authors state that fusing Endophilin's SH3 domain to UNC-26∆Sac did not rescue EPSC defects. Did the SH3 domain restore synaptic localization of ∆Sac? This would suggest that proper localization of the 5-Phosphatase domain is not sufficient to explain UNC-26 function in endocytosis*.

We added the Figure 9—figure supplement 1 to show that the SH3 domain enhanced the synaptic enrichment of UNC-26∆Sac1. We speculate that the lack of membrane interactions of the SH3 domain may explain why the SH3::UNC-26∆Sac1 construct fails to rescue. This is consistent with our results showing that the membrane binding activity of the BAR domain is required. Alternatively, the SH3 domain and the BAR domain may recognize distinct sub-synaptic regions at synapses. We added the following sentences to the manuscript:

“The functional difference between the BAR domain and the SH3 domain is likely due to their distinct binding partners and the potential for differential targeting to sub-synaptic regions.”

*5) In the same section of the manuscript, the authors speculate as to why the SH3 and PRD domains are conserved despite the fact that they are not required for SV endocytosis. Another possibility the authors may want to suggest is that the PRD/SH3 interaction provides a backup mechanism for co-localizing Endophilin and SJN under conditions where Sac1 localization is disrupted. For example, if Sac1 binds PI3,4P lipids, its localization may be disrupted when the levels of these lipids are low. In this case, the SH3/PRD interaction may be required for endocytosis*.

The reviewers raised an interesting point. We agree that while the PRD domain alone is not sufficient, it is possible that it become required in some situations, e.g., when the Sac1-membrane interactions are reduced. We have added the following statement in the Discussion section of the manuscript:

“Thus, it is possible that the PRD-SH3 interaction facilitates co-localization of endophilin and synaptojanin, helping the Sac1-dependent mechanism to sustain membrane recycling during persistent activity. In addition, while the PRD domain alone is not sufficient to promote SV endocytosis, it may become required in some situations, e.g., when the Sac1-membrane interactions are reduced.”

*6) In*
Figure 2*, what do the arrowheads indicate*?

We modified the legend of Figure 4 to clarify that “arrowheads indicate dense projections”.

*7) In*
Figure 7*, panel C, it is unclear what the cyan domain indicates in the BAR domain containing protein*.

The cyan domain indicates the SH3 domain of endophilin. We have modified the figure (now Figure 9).

*8) In*
Figure 7—figure supplement 1*, was rescue done with high copy or single copy transgenes*?

The rescue experiments in Figure 7—figure supplement 1 (now Figure 9—figure supplement 2) were done using high-copy transgenes.